# Carbon Capture Systems for Building-Level Heating Systems—A Socio-Economic and Environmental Evaluation

Don Rukmal Liyanage [1], Kasun Hewage [1,*], Hirushie Karunathilake [2], Gyan Chhipi-Shrestha [1] and Rehan Sadiq [1]

1  School of Engineering, Okanagan Campus, University of British Columbia, 1137 Alumni Avenue, Kelowna, BC V1V 1V7, Canada; liyanagedrd@alumni.ubc.ca (D.R.L.); gyan.shrestha@ubc.ca (G.C.-S.); rehan.sadiq@ubc.ca (R.S.)
2  Department of Mechanical Engineering, University of Moratuwa, Katubedda, Moratuwa 10400, Sri Lanka; hirushiek@uom.lk
*  Correspondence: kasun.hewage@ubc.ca; Tel.: +1-250-807-8176

**Abstract:** The energy consumption of buildings contributes significantly to global greenhouse gas (GHG) emissions. Energy use for space and water heating in buildings causes a major portion of these emissions. Natural gas (NG) is one of the dominant fuels used for building heating, emitting GHG emissions directly to the atmosphere. Many studies have been conducted on improving energy efficiency and using cleaner energy sources in buildings. However, implementing carbon capture, utilization, and storage (CCUS) on NG building heating systems is overlooked in the literature. CCUS technologies have proved their potential to reduce GHG emissions in fossil fuel power plants. However, their applicability for building-level applications has not been adequately established. A critical literature review was conducted to understand the feasibility and viability of adapting CCUS technologies to co-function in building heating systems. This study investigated the technical requirements, environmental and socio-economic impacts, and the drivers and barriers towards implementing building-level CCUS technologies. The findings indicated that implementing building-level CCUS technologies has significant overall benefits despite the marginal increase in energy consumption, operational costs, and capital costs. The information presented in this paper is valuable to academics, building owners and managers, innovators, investors, and policy makers involved in the clean energy sector.

**Keywords:** carbon capture; building heating; GHG emissions mitigation; techno-economic; triple bottom line sustainability

## 1. Introduction

The current phenomena of extreme weather, rising sea levels, and increases in droughts and floods indicate that the world is becoming more vulnerable to the ill effects of climate change [1]. Anthropogenic activities, such as fossil fuel combustion for energy generation that generates greenhouse gases (GHGs), have been identified as the dominant causes of increasing average global temperature levels and climate change [2]. Therefore, it is crucial to identify the most prominent GHG emitting economic sectors and investigate possible solutions to reduce GHG emissions.

The building sector is considered to be one of the most major energy consumers in the world [3]. Most of the building energy use is for heating purposes in colder climatic regions [3]. Fossil fuels such as natural gas, oil, and coal are the primary energy supply sources for building heating in cold climatic regions such as Canada [4]. However, coal-operated building heating systems are now very rare. This is mainly due to the availability of less expensive alternatives such as natural gas. In addition, coal combustion has adverse environmental and health impacts such as causing respiratory issues and

higher GHG emissions. Therefore, natural gas has become the most popular fossil fuel for building heating.

Building energy retrofits reduce energy consumption and associated operational GHG emissions from the building heating process in existing buildings. Building energy retrofits can be categorized as minor retrofits, such as upgrading the building envelopes; major retrofits, such as upgrading heating system efficiency; and deep retrofits, such as upgrading the heating system with renewable energy technologies [5]. Among the different types of retrofits, "deep" retrofits can limit operational GHG emissions by reducing energy demands and using cleaner energy sources.

Solar-thermal heating systems, ground-source heat pumps, and biomass energy systems are the main renewable energy technologies that are used for building heating [6]. However, these technologies are not commonly used in buildings due to various technical and economic limitations. Passive solar thermal systems are limited in their capacity to contribute towards building heating in the winter. Therefore, passive solar-thermal systems are not commonly used for building space heating in colder climates where the highest energy demands are in the winter [7]. Ground-source heat pumps can reduce energy demands considerably [8]. However, heat pumps require electricity for their operation. If electricity is generated using renewable energy, using ground-source heat pumps will significantly reduce operational GHG emissions [9]. Yet, around 66% of the world's electricity is generated using fossil fuels [10]. Therefore, using heat pumps will not be a solution for regions where electricity is primarily generated using fossil fuel. Biomass systems are meant to be a cleaner alternative to the fossil fuel supply. However, biomass heating systems are not commonly used for building heating due to the considerable biomass storage requirements and the challenges of developing an efficient logistic system to supply the required biomass [11]. In addition to all of the above technical factors, significantly high investment costs are also a barrier to integrating renewable energy technologies [12].

In addition to the above-mentioned energy efficiency and renewable energy interventions, carbon capture, utilization, and storage technology (CCUS) is becoming an emerging alternative for mitigating the emissions associated with fossil fuels. CCUS technology separates $CO_2$ from combustion sources such as chemical industries and fossil fuel power plants. The captured $CO_2$ is stored in geological formations or is utilized in usable products [13]. This approach is commercially used in inherent $CO_2$ separation applications such as in NG processing and chemical production, which produce high-density $CO_2$ [14]. Recently, the prospects of downsizing the existing carbon capture strategies to reduce GHG emissions from buildings have been considered. The potential of emission reduction without compromising building energy economics is the main motivation behind this strategy. Some pilot-scale carbon capture devices have been developed for use in NG building heating systems [15]. However, the lack of economic, environmental, and social assessments (triple bottom line of sustainability assessment) as well as a lack of feasibility assessment and research and development activities are critical challenges for the successful commercialization and market penetration of building-level carbon capture systems.

Several studies reviewing the CCUS technology literature can be found. Rosa M. and Azapagic A. conducted a critical review on the life cycle environmental impacts of CCUS technologies [16]. González-Salazar reviewed recent developments in the carbon capture technologies used in gas power generation [17]. Asif M. et al. and Vega F. et al. reviewed the current status of the chemical absorption technologies used for carbon capture [18,19]. These literature reviews identified challenges and prospects of implementing CCUS technologies in the fossil fuel power generation sector and of scaling up the deployment of CCUS technologies. In addition, Hetti R. et al. conducted a literature review on integrating CCUS technologies into community-scale energy systems [20]. The authors considered the prospects of downsizing the CCUS technologies used in large-scale fossil fuel power generation plants into centralized community energy systems [20]. The study scope was limited to community-scale electricity generation plants and district energy systems [20]. However, there is a lack of information on the prospects and challenges of implementing

CCUS technologies in building-scale heating systems in those studies. Specifically, technical requirements, environmental and socio-economic impacts, and the drivers and barriers of building-level CCUS technologies have yet to be explored.

This paper aims to investigate the potential of reducing the GHG emissions of NG building heating systems by implementing carbon capturing, storage, and utilization (CCUS) technologies. This study critically reviews published articles on CCUS technologies used in fossil fuel power generation and discusses the possibility of adopting all of the possible carbon capturing process stages in the building context. This study consists of three main sections. The first section provides a brief overview of the CCUS technologies used in fossil fuel power generation. The second section discusses the technical adaptation of carbon capture in building-level applications. Finally, the sustainable implementation of carbon capture at the building level is discussed by considering environmental, economic, and social aspects. The compilation of the information found here is useful for researchers and innovators to study the technical feasibility and triple bottom line sustainability of implementing carbon capture at the building level.

## 2. Materials and Methods

Keyword searching in subject-specific databases such as "Compendex Engineering Village" and "ScienceDirect" was used to collect the relevant literature. The study used the keyword combinations of "carbon capture", "storage", utilization", "building heating", and "post-combustion" to search for studies from databases mentioned above. The study prioritized 51 journal articles published after 2005 from 13 high impact factor journals (with impact factors above 2.5). The selected journals and their impact factors are listed in the Table 1 shown below.

**Table 1.** Primary sources of published literature.

| Journal Paper | Impact Factor (2018) | Number of Papers |
|---|---|---|
| Renewable and Sustainable Energy Reviews | 10.556 | 5 |
| Aerosol and Air Quality Research | 2.735 | 1 |
| Applied Energy | 8.426 | 8 |
| Applied Thermal Engineering | 4.026 | 1 |
| Chemical Engineering Science | 3.372 | 1 |
| Energy | 5.537 | 1 |
| Energy Conversion and Management | 7.181 | 3 |
| Energy Policy | 4.88 | 2 |
| Fuel | 5.128 | 2 |
| International Journal of Greenhouse Gas Control | 3.231 | 15 |
| Journal of Cleaner Production | 6.395 | 4 |
| Journal of $CO_2$ Utilization | 5.844 | 2 |
| Journal of Membrane Science | 7.015 | 2 |

Apart from that, publications published prior to 2005 were used in cases where more recent information was unavailable. Furthermore, Canadian Government reports, conference proceedings, relevant websites, and other reports related to carbon capture and building heating were also considered.

## 3. An Overview: Carbon Capture, Utilization, and Storage (CCUS) Technologies

According to the International Energy Agency, 30 million tons of $CO_2$ are captured annually by carbo capture facilities. Out of those 30 million tons, 90% is captured from oil and gas production industries, which produce high-density $CO_2$ streams [21]. Technologies for capturing high-density $CO_2$ have been widely deployed and have reached technological maturity. However, most stationary combustion sources produce low concentration $CO_2$, and the technologies used to capture $CO_2$ from these sources are in the initial deployment stage.

The carbon capture technologies used for stationary combustion energy sources can be classified as post-combustion, pre-combustion, and oxy-fuel combustion technologies [22]. This classification is based on the combustion process and gas extraction point. Post-combustion carbon capture technology is used to capture $CO_2$ from flue gas after combustion is completed [23]. It can be used to capture $CO_2$ from fossil fuel power plants [14,24,25], process heaters, and combined heat and power plants used in chemical production facilities. This technology has been identified as the most practical carbon capture technology, as it can be implemented as a retrofit to existing stationary combustion sources without changing the infrastructure and the combustion method considerably [22,26,27]. Post-combustion carbon capture is considered the most mature carbon capture technology in the power generation sector and is in the early stages of deployment [28].

Pre-combustion technology is used to capture $CO_2$ from the fuel before the combustion process begins [22]. The pre-combustion capture process is generally used in fuel gasification processes, where coal [22], biomass [29], or natural gas [25] is used as the primary fuel. It is in the early stages of deployment and commercializing projects [28]. In oxy-fuel combustion, fuel is reacted with pure $O_2$ diluted with recirculated flue gas. The oxygen is separated from the air by means of the cryogenic separation method [30]. However, large-scale oxy-fuel carbon capture facilities have not been established due to the high energy requirements needed for the $O_2$ separation [22,31].

Post-combustion and pre-combustion technologies require carbon separation methods to separate $CO_2$ from gas. Approaches to separate $CO_2$, such as absorption, adsorption, and membrane separation, are well known in the industry. In contrast, the oxy-fuel method does not require any specific $CO_2$ separation method, as the combustion products are only $CO_2$ and water vapor. The water vapor can be removed through the condensation of the combustion products [32]. The captured $CO_2$ has to be stored or utilized to stop $CO_2$ from being released into the atmosphere. Moreover, it has to be compressed and liquefied after the capturing process depending on the $CO_2$ transportation, storage, and utilization method [33]. Figure 1 shows the carbon capture, storage, and utilization process of post-combustion and pre-combustion technologies.

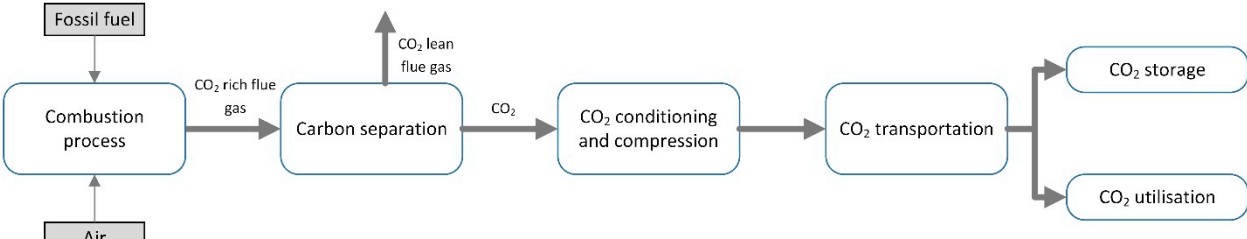

**Figure 1.** Post-combustion carbon capture process.

### 3.1. Carbon Separation Methods

The main modes of separation that are currently in practice are absorption, adsorption, chemical looping, membrane separation, hydrate-based separation, and cryogenic distillation. In the absorption process, the $CO_2$ from the flue gas is absorbed by a liquid solution called an absorbent [34]. Chemical absorption and physical absorption are the main processes [31,34]. In chemical absorption, $CO_2$ reacts with the chemical solvent and forms an intermediate compound [35]. In physical absorption, the $CO_2$ bonds with the solvent using Van der Waals forces in a liquid solution without any reaction [36]. Generally, the bonds formed between $CO_2$ and the solvent in chemical absorption are stronger than the bonds formed in physical absorption. Therefore, the $CO_2$ absorption efficiency in chemical absorption is higher than that of physical absorption. Chemical absorption is more suitable for capturing $CO_2$ from flue gas with low pressure and a low $CO_2$ concentration [35]. Chemical absorption is used in post-combustion technology, while the physical absorption method is used in pre-combustion technology [37,38].

In the adsorption method, the substances (adsorbate) adhere to a solid surface (adsorbent). The adhered substances can be removed later by changing the temperature or pressure. The adsorption process can be categorized as physisorption and chemisorption. The adsorption and desorption processes are performed using three main methods: pressure swing adsorption, vacuum swing adsorption, and temperature swing absorption. Apart from that, electric swing adsorption and pressure and temperature hybrid processes are also used for the adsorption process, which are considered to be advanced technologies [39]. Furthermore, the adsorption method can be used for post-combustion capture [16].

Membrane separation is a novel technology compared to the other separation methods discussed above. This carbon separation method is considered to be a flexible method, as it can be used in post- and pre-combustion technologies [40]. In membrane technologies, most of the energy is consumed in order to develop the required pressure difference across the membranes [41]. This technology is very economical when high-purity $CO_2$ is not required. Post-combustion technology requires membranes with high selectivity, as the $CO_2$ concentration of the flue gas is very low [40]. Membrane systems that have high selectivity consume more energy and have significantly higher costs compared to low-selectivity membranes. Therefore, it is challenging to implement membrane systems commercially in post-combustion carbon capture systems [42], leading to membrane separation methods still only being implemented at lab scale. Figure 2 shows the classification of carbon capture technologies.

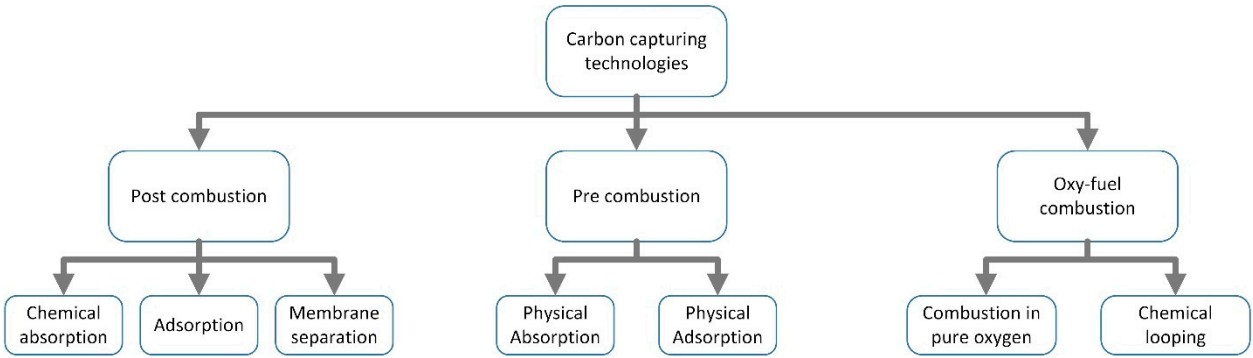

**Figure 2.** Classification of carbon capture technologies.

### 3.2. $CO_2$ Transportation

The captured $CO_2$ should be transported from the carbon sources to specific storage locations. In the USA, several million tons of $CO_2$ are transported annually for enhanced oil recovery. The transportation methods should be capable of transporting $CO_2$ efficiently with little leakage. More importantly, the transportation methods should be economically viable [43]. The $CO_2$ can be transported using pipelines, tanker trucks, ships, and railroads [44]. Pipeline systems are the most efficient and viable method used to transport $CO_2$ on a large scale. Tanker trucks are used to transport $CO_2$ in the short term and over short distances. Moreover, transporting $CO_2$ using tanker trucks and railroads is overlooked in the literature.

Pipeline transportation can be used for both onshore and offshore $CO_2$ transportation [45]. However, pipelines are not tested for offshore $CO_2$ transportation [46]. Fixed or towed pipes are the most commercially viable method to transport $CO_2$ to the ocean. Recompression stations are used to reduce the pressure head (i.e., compensate for the pressure head). Pipeline transportation facilities consists of a $CO_2$ conditioning facility that conducts $CO_2$ compression and further separation from water vapor and other gases. Generally, $CO_2$ is compressed to 100–150 bar in order to transport the $CO_2$ through pipe lines [45]. In some cases, $CO_2$ is compressed to the liquid state so that it can be pumped. This method reduces the energy requirement for transporting $CO_2$ through pipelines [45].

Waterborne transport is another method that can be used to transport $CO_2$ over very large distances [47]. Ships and other types of watercrafts can be used to transport $CO_2$ under conditions where pipelines are not viable. The $CO_2$ should be in liquid form in order to reduce the transport volume when using ships [48]. In contrast to $CO_2$ transport using pipelines, ship transportation is a discrete transportation mode [48]. Therefore, ship transportation requires buffer storage or temporary storage [49,50]. The $CO_2$ is transferred from the temporary storage site to the ship loading facilities. Ship transport can be used to transport $CO_2$ from the loading facilities to offshore or onshore storage facilities. Furthermore, studies are being conducted where $CO_2$ is injected directly into the ocean using ships [50].

### 3.3. Carbon Storage

The captured $CO_2$ can be stored in geological storage and in offshore storage and can be converted into mineral carbonates [16]. Depleted oil or natural gas reservoirs [51] and saline aquifers are geological $CO_2$ storages. Furthermore, un-mineable coal beds are also considered to be geological $CO_2$ storage. Generally, $CO_2$ is injected into geological formations at depths higher than 800 m [16]. Geological storage should consist of a porous rock and cap rock to store $CO_2$. The porous rock acts as the storage medium where $CO_2$ is stored. The caprock is used to avoid $CO_2$ leakage from the storage. The $CO_2$ is trapped in the storage site by means of physical trapping, dissolution in saltwater, and absorption into coal or organic-rich shale replacing methane ($CH_4$) and other gases. The dissolved $CO_2$ can be reacted with rocks and minerals and can be stored permanently. The $CO_2$ can be stored as compressed gas, liquid $CO_2$, or in the supercritical phase. This choice depends on the storage conditions [16]. Furthermore, storing $CO_2$ in geological formations has become a promising option due to the oil and gas industry's expertise in geological formations [16].

The ocean is a natural carbon sink that currently absorbs 7 $GtCO_2$ per year [47]. Apart from that, $CO_2$ can be intentionally injected into the sea using the ocean storage method [52]. Here, $CO_2$ is injected into the water column of the ocean or the seafloor. It is possible to inject $CO_2$ into the sea in the form of gases, liquids, solids, and hydrates, depending on the injection technology. The $CO_2$ is dissolved in the ocean regardless of the form it is injected in. It has to be injected at a depth less of than 500 m to release $CO_2$ as gas. When $CO_2$ is released below 500 m and above 2500 m, it is released as a liquid and moves upward (towards the surface of the water) while dissolving. If the release depth is higher than 2500 m, the $CO_2$ is released as a liquid and moves down (towards the ocean floor). It is possible for the $CO_2$ to be dissolved completely before it arrives at the ocean surface or to remain as a $CO_2$ lake at the ocean floor until it is completely dissolved. Deep ocean storage is still in the research phase, and there are no pilot-scale projects that are currently ongoing [47].

### 3.4. $CO_2$ Utilization

There are various methods of using $CO_2$ in industry. The carbon capture can be used as a chemical feedstock in industries and can used in applications such as synthesizing methanol and other types of polymers [53]. Furthermore, it can be used directly as a carbonating agent, preservative, and solvent in the food and beverage industry [16,54]. Moreover, $CO_2$ is used as a working fluid in refrigeration cycles [55]. In addition, $CO_2$ is used in many industries, such as steel manufacturing, power generation, metalworking and welding, and pneumatics [54].

Enhanced oil recovery (EOR) is also a fuel production method with an increasing demand for $CO_2$ [16]. In the EOR process, $CO_2$ is injected with other chemicals into an underground oil reservoir in order to remove the oil trapped in the rocks [56]. This method can extract more than 30–60% of the trapped oil [16]. Most $CO_2$ is removed along with the oil, and the oil needs to be treated before use. However, some of the $CO_2$ may be released into the atmosphere during this treatment process [16].

In addition, other industrial uses have been introduced in recent times. Mineral carbonation is considered to be a storage method in few studies [16,57,58], while others consider it to be a utilization method [16,54]. In this process, $CO_2$ is reacted with minerals such as Wollastonite and Serpentine and form mineral carbonates [57]. Therefore, $CO_2$ can be permanently stored within a chemical. On the other hand, this method has a higher capacity than all other fossil reserves, as magnesium and calcium-rich minerals can be efficiently mined [58]. However, this method requires input energy, thus in directly contributing to additional GHG emissions.

Bonaventura et al. (2017) described a novel method of capturing $CO_2$. This process produces $NaHCO_3$ as a by-product during the carbon capture process [26]. This process uses Trona as the chemical solvent, which is a low-cost mineral used to produce $Na_2CO_3$. The process can be controlled so that only a fraction of the $CO_2$ is utilized. The other fraction can be stored or utilized in another method. It is also possible to use ammonia ($NH_3$) to capture $CO_2$ while producing ammonium salts [59]. In this process, the ammonium salts have to be separated from the solvent using filtration or sedimentation. The separated ammonium can be used in the agriculture industry as a fertilizer ingredient [60]. Utilizing $CO_2$ in another product may help to avoid energy consumption and GHG emissions related to the production of that product.

A summary of the carbon storage and utilization applications discussed above is shown below in Figure 3.

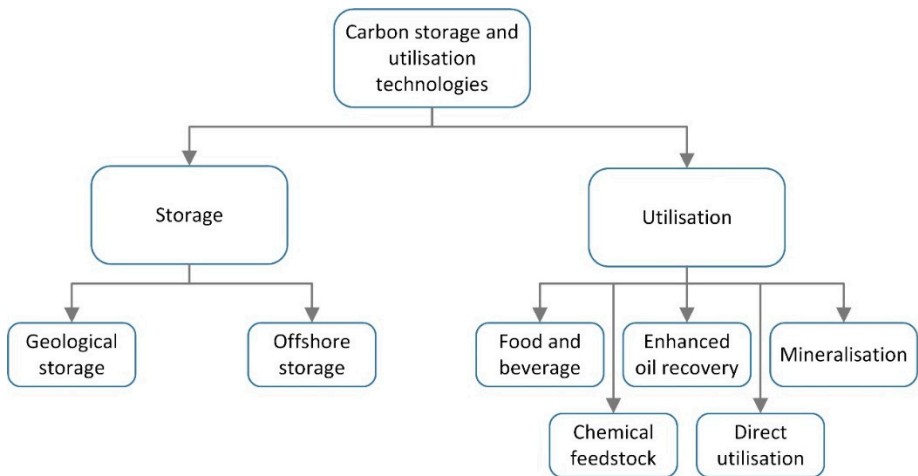

**Figure 3.** Classification of carbon storage and utilization methods.

## 4. Adoption of the CCUS Value Chain for Building Scale

Liyanage [61,62] and Pokhrel [63] mentioned a carbon technology that can be integrated into building heating systems. Based on the literature review, this technology was the only one that was found to be used in building-scale applications. The system uses solid potassium hydroxide (KOH) to capture the $CO_2$ from the exhaust gas emitted from natural gas building heating systems. The reaction between KOH and $CO_2$ produces potassium carbonate ($K_2CO_3$) as a by-product. The chemical reaction is given in Equation (1) below. The by-product is widely used for pharmaceutical purposes, soap production, and as a chemical feedstock in many industries [63].

$$2KOH(s) + CO_2(g) \rightarrow K_2CO_3(s) + H_2O(l), \qquad (1)$$

In addition, the chemical reaction given in Equation (1) is exothermic and therefore generates heat during the carbon capture process. The carbon capture system recovers heat from the chemical reaction and the waste heat from the flue gas. The recovered heat is transferred into the domestic hot water system. This carbon capture system is currently available commercially and is only used in commercial buildings. Liyanage [61,62] con-

ducted experimentation on the above-mentioned carbon capture system and found that its average carbon recovery rate is 13%. In addition, the average heat transfer rate from the heat recovery system to the domestic hot water system is 26 kJ/l. Currently, the system can contain 200 kg of KOH per month [61,62]. Hereafter, this technology will be referred to as KOH-based building-level carbon capturing technology.

It is also important to consider the adoptability of the other CCUS technologies used in the power generation sector at the building level. The CCUS value chain consists of carbon capture, $CO_2$ transportation, and $CO_2$ storage or utilization. All of the processes involved in CCUS technology must be successfully adapted in order for carbon capture technology to be used at a building scale. Among the three carbon capture technologies discussed about, pre-combustion capture technology cannot be used with natural gas building heating furnaces, as there is no intermediate $CO_2$ generation during the combustion process. The oxy-fuel combustion method needs an additional oxygen supply and a different combustion system and therefore is not considered in this review. Only post-combustion technology shows promise in investigating the potential of adopting the CCUS value chain at the building scale.

### 4.1. Operational Conditions and $CO_2$ Output of Carbon Separation Technologies

Chemical absorption, adsorption, and membrane separation are the separation technologies that are used in post-combustion carbon capture technology, as explained in Section 3.1. Flue gas properties, including temperature, pressure, and $CO_2$ concentration, are considered important parameters when selecting suitable carbon separation technologies [64]. Table 2 shows the operational conditions of the above-mentioned carbon separating technologies. In addition, Table 2 shows the optimum $CO_2$ composition and the $CO_2$ purity after the separation process in the chemical absorption, adsorption, and membrane separation methods [65].

**Table 2.** Operating conditions and outputs of carbon separation technologies [34,65–68].

| Carbon Separation Method | Operating Temperature | $CO_2$ Composition | $CO_2$ Purity | $CO_2$ Capture % |
|---|---|---|---|---|
| Chemical absorption using Methyl Ethanolamine (MEA) | 45–50 °C | >5% | >95% | 80–95% |
| Chemical absorption using Econamine | 80–120 °C | >5% | >95% | 80–95% |
| Chemical absorption using 2n Methyl Diethanolamine (MDEA) | 35–40 °C | >5% | >95% | 80–95% |
| Chemical adsorption PSR | 50 °C | >10% | 75–90% | 80–95% |
| Physical adsorption PSA | 50–100 °C | >10% | 75–90% | 80–95% |
| Membrane separation | - | >15% | 80–95% | 60–80% |

The $CO_2$ composition of flue gas from a natural gas combustion systems varies from 7% to 10% [50]. Therefore, chemical absorption technology must be used with building heating systems without making any modifications to the boiler system. Membrane separation and adsorption processes cannot be used directly with natural gas building heating systems. The optimum $CO_2$ composition in membrane separation is higher than that found in the flue gas in natural gas heating systems [40]. However, recent studies indicate that the $CO_2$ composition of the flue gas can be increased by recirculating the flue gas through the combustion system [50]. This procedure is used in NGCC combustion systems, as the $CO_2$ composition of flue gas is 3–4% [50]. The same procedure can be used in building heating systems after some modifications to the combustion process.

The temperature of the flue gas reduces with the increasing efficiency of the heating system. Annual fuel utilization efficiency (AFUE) categorizes building heating systems based on their efficiency. A standard efficient building has the lowest possible AFUE, which is 78–80%, and the flue gas temperature in these buildings is approximately 232 °C [69]. Mid-efficiency furnaces are widely used in buildings, and the efficiency can reach 83% with a flue gas temperature of 149 °C [69]. Therefore, flue gas must be cooled in both standard

and mid-efficiency furnaces. High-efficiency condensing heating systems emit flue gas at a much lower temperature, approximately 50 °C [69]. This indicates that highly efficient furnaces can be used without cooling systems in most of the carbon separation technologies shown in Table 2.

*4.2. Energy Consumption of the Carbon Separation Process*

The carbon capture process requires energy for its operation. Chemical absorption technology requires thermal energy for solvent regeneration. In addition, electricity is required for the operation of auxiliary equipment such as pumps and compressors. Table 3 shows the energy consumption of the chemical absorption method with different types of solvents.

**Table 3.** Energy consumption of chemical absorption technology [70–74].

| Separation Process | Desorption Energy | Auxiliary Energy |
|---|---|---|
| Commercial Level Solvents | | |
| Chemical absorption with MEA | $3.53 \, GJ_{th}/tCO_2$ | $0.0432 \, GJ_e/tCO_2$ |
| Chemical absorption Econmaine FG+ | $3.18 \, GJ_{th}/tCO_2$ | - |
| Chemical absorption KS-1 | $3.08 \, GJ_{th}/tCO_2$ | - |
| Chemical absorption KS-2 | $3.0 \, GJ_{th}/tCO_2$ | - |
| Chemical absorption CANSOLV | $2.33 \, GJ_{th}/tCO_2$ | - |
| Chemical absorption H3 | $2.8 \, GJ_{th}/tCO_2$ | - |
| Chemical absorption with UNO MK3 | $2.24 \, GJ_{th}/tCO_2$ | $0.0612 \, GJ_e/tCO_2$ |

In the chemical absorption method, the required energy has to be supplied as heat through steam. The temperature of the steam should be in a range from 100 °C to 140 °C [75]. Generally, the steam is extracted from steam turbines in power plants that are integrated with carbon capture systems. Therefore, there is a possibility of using thermal energy from standard-efficiency furnaces, as the temperature of the flue gas from these furnaces is 232 °C. In addition, using the thermal energy from low efficiency furnaces may require separate cooling systems to reduce the flue gas temperature. Medium- and high-efficiency furnaces must be modified to extract thermal energy, as the flue gas temperature is low. However, these furnaces might reduce the heat generation of the furnace. As a solution, the required thermal energy can be supplied using electric heaters. Some studies have been conducted on integrating solar energy systems for carbon capture systems to reduce the regeneration energy requirement from the power plant [75,76]. The same procedure can be applied to building-level heating systems that have been integrated with carbon capture systems to minimize fuel consumption. Furthermore, chemical absorption technology requires energy to operate compressors, pumps, condensers, and re-boilers, which are the auxiliary components of carbon capture systems.

Adsorption and membrane separation methods do not need thermal energy for their operation. Instead, these technologies require electricity for the compression, vacuum generation, and running of the auxiliary components. The post-combustion technology requires membranes with high selectivity, as the $CO_2$ concentration of the flue gas is very low [40]. Membrane systems with high selectivity consume more energy and are significantly more costly than low-selectivity membranes. Therefore, it is challenging to implement the membrane systems in post-combustion carbon capture systems commercially [38], and these systems are not commonly used commercially in natural gas power plants. Table 4 shows the energy consumption of adsorption and membrane separation technologies.

**Table 4.** Energy consumption of adsorption and membrane separation technologies [71,72].

| Separation Process | Energy Requirement |
|---|---|
| VPSA | 2.140 $GJ_E/tCO_2$ |
| PSA | 2.3–2.8 $GJ_E/tCO_2$ |
| TSA | 6.12–6.46 $GJ_E/tCO_2$ |
| Membrane separation POL-POL | 0.5–6 $GJ_E/tCO_2$ |

*4.3. Operational Parameters of Carbon Separation Technologies*

A. Brunetti et al. investigated the major operational parameters that affect the efficiency of carbon capturing systems [74]. The authors mentioned that operational flexibility, turndown, and reliability are important parameters when designing carbon capture systems. The definitions of the above operational parameters are shown below.

- Operational flexibility: The ability of the system to operate in variable gas compositions [65];
- Turndown: The ability of the system to operate under gas flow rates that are less than the design flow rates [65];
- Reliability: The ability to operate continuously without unscheduled shutdown [65];
- Adaptability: The time required to adapt the carbon capture system for the changes in the inflow properties [65].

Brunetti et al. showed that membrane systems are highly flexible when the $CO_2$ concentration is higher than 20% [74]. The flexibility of membrane systems is reduced dramatically when the $CO_2$ concentration is less than 20%. As a result of the composition changes, the $CO_2$ recovery rate and the purity of $CO_2$ are reduced. The adsorption method is also considered to be a highly flexible carbon capture technology. The absorption systems are moderately flexible compared to the membrane systems. In addition, the absorption systems require changes in the liquid flow rate when the gas composition changes [77]. The liquid flow rate is limited by the size of the systems and thus restricts the flexibility of the absorption system. This indicates that the absorption systems must be oversized when the systems are subjected to higher gas composition variations. Brunetti et al. showed that membrane systems are highly flexible when the $CO_2$ concentration is higher than 20% [74]. The flexibility of membrane systems is reduced dramatically when the $CO_2$ concentration is less than 20%. As a result of these composition changes, the $CO_2$ recovery rate and the purity of $CO_2$ are reduced. The adsorption method is also considered to be a highly flexible carbon capture technology. The absorption systems are moderately flexible compared to membrane systems. In addition, absorption systems require changes in the liquid flow rate when the gas composition is changed [77]. The liquid flow rate is limited by the size of the systems and thus restricts the flexibility of the absorption system. This indicates that the absorption systems must be oversized when the systems are subjected to higher gas composition variations.

Most power generation plants are operated with a steady combustion rate. In contrast, the thermal energy load of the building changes considerably over time. As a result, the fossil fuel combustion rate and the $CO_2$ flow rate change. Therefore, carbon capture systems must be able to maintain their performance regardless of the variations in the gas flow rates. Therefore, the turndown capability of carbon capture systems is important when used in the building context. A. Brunetti et al. showed that membrane systems can maintain the purity of the $CO_2$ stream even at 10% of the design flow [74]. Therefore, membrane systems can be defined as systems with a higher turndown capability. Absorption technology can maintain its $CO_2$ recovery and $CO_2$ purity downstream at 30 to 100% of its design flow. Although chemical absorption technology can maintain purity even when the flow is less than 30% of its design flow, $CO_2$ recovery can be reduced considerably. Adsorption technology can also deliver the expected $CO_2$ recovery and $CO_2$ purity, even at 30% of its design flow [65,78].

Although operational flexibility and turndown measure the resilience of carbon capture systems for variations in the flow and composition, it is essential to investigate how much time is needed for system adaptation. Building heating systems especially are subjected to frequent load changes. Membrane separation systems can adapt to such variations instantaneously, while absorption and adsorption technologies can adapt within 5–15 min.

A building environment has less technical experts than the industrial environment does, which is where carbon capture systems are currently installed. Therefore, carbon capture systems must be more reliable. Membrane separation is known to be extremely reliable, as it has less control components [74]. The absorption method is considered to be moderately reliable [65,74]. More specifically, the equipment used to reduce chemical degradation can cause unscheduled shutdowns and may require frequency maintenance. The adsorption method is also moderately reliable [65] compared to the membrane separation method. A summary of the carbon separation technologies that are currently being adopted is given in Table 5.

**Table 5.** Summary of carbon separation technologies adopted at building level.

| Technical Criteria | Chemical Absorption | Adsorption | Membrane Separation | Suitability of the Carbon Capture System |
|---|---|---|---|---|
| | | *Operational Conditions* | | |
| *Operating temperature* | Gas cooling is required for standard- and medium-efficiency furnaces Gas cooling is not required for high-efficiency furnaces | Gas cooling is required for standard- and medium-efficiency furnaces Gas cooling is not required for high-efficiency furnaces | Gas cooling is required for standard- and medium-efficiency furnaces Gas cooling is not required for high-efficiency furnaces | Absorption, adsorption, and membrane separation have the same suitability |
| *$CO_2$ composition* | Exhaust gas recirculation is not needed Best option | Exhaust gas recirculation is needed Moderate option | Exhaust gas recirculation is needed | Absorption method is more suitable |
| | | *Carbon capture performance* | | |
| *$CO_2$ purity* | Captured $CO_2$ can be used with any utilization and storage method | Captured $CO_2$ can be used with few utilization and storage methods | Captured $CO_2$ can be used few utilization and storage methods | Absorption method is more suitable |
| *$CO_2$ capture rate* | Has higher $CO_2$ recovery | Has moderate $CO_2$ recovery | Has low $CO_2$ recovery | Absorption method is more suitable |
| *Energy requirement* | Thermal energy is required—The furnace can be modified or can use electrical heating Medium energy requirement | Only electricity is needed. High energy requirement | Only electricity is needed. High energy requirement as the $CO_2$ concentration is low | Absorption method is more suitable |
| | | *Operational parameters* | | |
| *Operational flexibility* | Medium flexibility | High flexibility | Low flexibility (For $CO_2 < 20\%$) | Adsorption method is more suitable |
| *Turndown* | Can maintain $CO_2$ recovery and purity down to 30% Can maintain $CO_2$ purity below 30% | Can maintain $CO_2$ recovery and purity down to 30% | Can maintain $CO_2$ recovery and purity down to 10% | Membrane separation method is more suitable |
| *Reliability* | Medium reliability | Medium reliability | High reliability | Membrane separation is more suitable |
| *Adaptability* | Within 5–15 min | Within 5–15 min | Instantaneous | Membrane separation is more suitable |

Membrane separation technology shows higher performance with regard to operational flexibility, turndown, adaptability, and reliability compared to other technologies. However, membrane separation requires a higher percentage of $CO_2$ in the inflow (over 20%), which is considerably higher than that of the flue gas composition (less than 10%) of natural gas building heating systems. Although flue gas recirculation is a possible solution [79,80], it may need considerable modifications in the existing heating systems that require further research. Adsorption technology has a lower performance compared to all of the above factors. It performs well in terms of operational flexibility compared to absorption technology, although absorption technology requires less energy. However, adsorption technology may also require flue gas recirculation since it operates at a higher $CO_2$

percentage (>10%). The absorption technology has moderate performance and considers all of the factors while working at a very low $CO_2$ concentration (>5%). Thus, absorption technology may be more applicable for natural gas furnaces, as it does not require any modifications to the building heating system.

### 4.4. Transportation of CO₂ and By-Products in Building Scale

The $CO_2$ utilization and storage step can be a critical phase in building-level carbon-capturing that defines the economic viability of the whole process. It is necessary to transport $CO_2$ over very long distances so that it can be stored within geological storage sites, especially in Canada, where geological carbon storage is widely dispersed. Pipeline transportation is the only commercially available method to transport $CO_2$ over long distances [81]. However, carbon transportation using pipelines from individual buildings would not be economically viable. Considerable capital investment is required in developing such infrastructure. Middleton and Bielicki (2009) showed that pipeline transportation costs would be extremely high for low $CO_2$ flow rates [82]. Moreover, $CO_2$ needs to be highly compressed and conditioned for transportation, which increases the costs for small-scale applications considerably. Therefore, $CO_2$ transport using pipelines and storage within geological storage sites would not be economically viable for small-scale applications such as building heating systems.

Road transportation is a less attractive option in large-scale $CO_2$ transportation applications. Road transportation costs twice as much as pipeline transportation in large-scale carbon capture and storage projects [83]. However, tanker trucks are used to transport $CO_2$ from $CO_2$ distribution terminals to customers for carbon utilization purposes [83]. Generally, $CO_2$ should be in liquid form when being transported by tanker trucks to maximize transportation capacity. Therefore, compression and refrigeration systems must be integrated into carbon capture systems. In addition, intermediate storage systems must be implemented in buildings where $CO_2$ is stored. When $CO_2$ is converted into a by-product during the carbon capture process [26,59], the by-products must be transported instead of the $CO_2$ gas. This reduces the space requirement as well as the energy requirement for $CO_2$ compression and liquefaction. However, this process requires the frequent loading and unloading of chemicals. Therefore, the public acceptance of utilizing carbon during the separation process would be questionable.

Buildings in colder climatic regions produce significant amounts of $CO_2$ emissions due to the higher thermal energy requirements. For example, an average residential building in Canada that uses a NG heating system emits approximately 6 tons of $CO_2$ per year [84]. Furthermore, most of these emissions are generated during the winter season and may exceed 1 ton of $CO_2$ per month for an average residence. Therefore, the carbon capture percentage is mostly limited by material handling and transporting capacity despite the higher $CO_2$ capture efficiency of modern carbon capture technologies. Therefore, the viability of building-level carbon capture systems mainly depends on efficient $CO_2$ and by-product transporting methods.

### 4.5. Technical Drivers and Barriers

The above review revealed that chemical absorption technology is suitable for operation in building-level heating systems based on flue gas properties without necessitating substantial changes to the combustion system. However, the chemical absorption method requires 5–15 min to adapt to the changes in flue gas rates. In addition, it only has moderate reliability. Therefore, chemical absorption technology may require substantial R&D efforts to improve the control mechanisms and reliability in order to integrated with building-level heating systems. Membrane separation, which is more favorable for building operations under most criteria, requires flue gas circulation due to the lower $CO_2$ concentrations. Therefore, heating systems may require considerable modifications in order to be used with membrane separation technologies. In addition, buildings have limited space compared to power generation plants. Therefore, one of the main barriers to implementation in

buildings is space limitations. Specifically, chemical absorption requires taller columns that may not be feasible for installation in buildings [85], while membrane separation may need a large area [86].

The transportation of the captured $CO_2$ or by-products is also one of the main challenges. Although pipeline transportation is commonly used in large-scale facilities, using it at the building scale may not be practical due to large infrastructure requirements. Road transportation would be the most practical method for building-level applications although it is a discrete mode of transportation [47]. Road transportation requires intermittent $CO_2$ storage in buildings, which may require considerable space. In addition, the captured $CO_2$ must be liquefied to be stored and transported, which requires a considerable amount of energy. Therefore, the potential for $CO_2$ reduction is restricted by the available space in the building when using road transportation. Figures 4 and 5 show possible pathways to implement building-level carbon capture technologies that were identified based on the literature review.

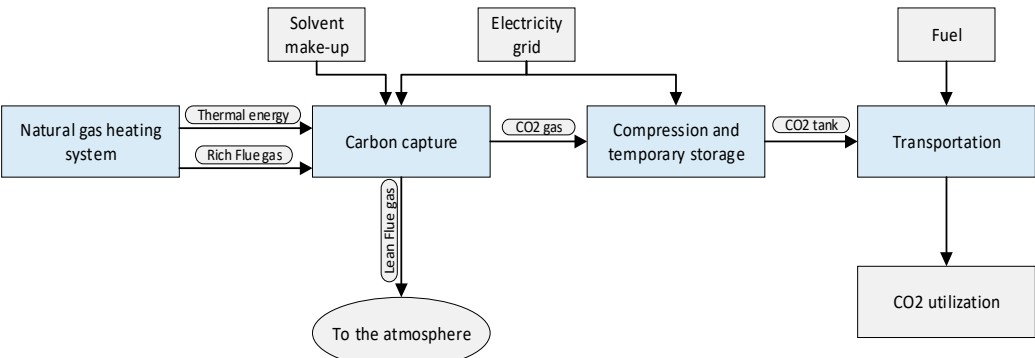

**Figure 4.** The proposed building-level carbon capture process that separates $CO_2$ from flue gas (Ex: MEA-based chemical absorption).

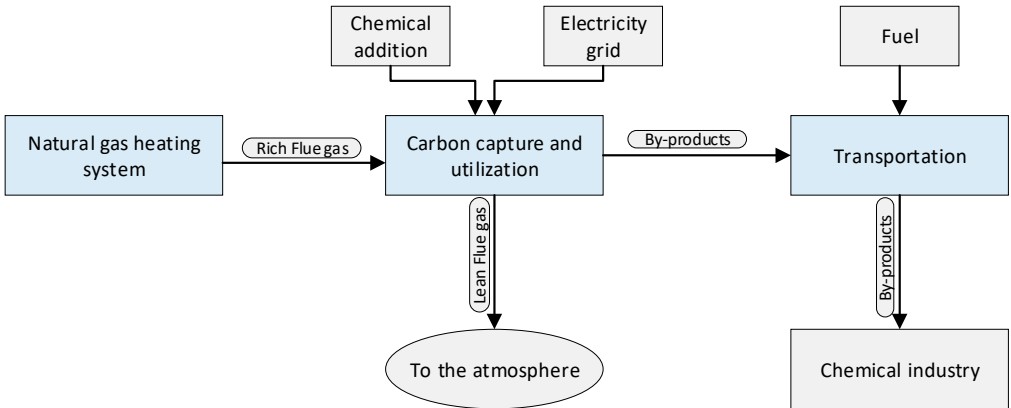

**Figure 5.** The carbon capture process that converts $CO_2$ into a by-product (KOH-based building-level carbon capture technology).

## 5. Implementation of Carbon Capture, Storage, and Utilization at Building Level

This study has revealed that integrating carbon capture has potential when considering the technical aspects. However, an increase in energy demand, emissions during the operation, and the production of the raw material required for the carbon capture process may cause significant environmental impacts in the life cycle of the carbon capture process. Furthermore, the carbon capture process also carries substantial economic burdens by increasing energy, material, transportation costs. In addition, the carbon capture process may increase the maintenance work required in building heating systems, which can reduce the acceptability of implementing carbon capture to the building owners. Therefore, it is necessary to consider the environmental impacts, economic costs and benefits, and

social acceptance of adapting the carbon capture value chain to building-level heating systems successfully.

*5.1. Environmental Impacts*

The carbon capture life cycle consists of material acquisition, carbon capture and storage facility construction, development of the carbon capture phases, $CO_2$ transportation, and $CO_2$ storage and utilization. Each stage consists of various material and energy flows that affect the overall life cycle performance of the carbon capture strategy. The life cycle assessment (LCA) method is commonly used to assess the performance of the whole process and to observe the holistic impact of a system.

Singh, Strømman, and Hertwich (2011) conducted an LCA on natural gas combined cycle power plants with MEA-chemical absorption carbon capture and storage [87]. The study shows that the MEA chemical absorption method can reduce total GHG emissions by 75%. However, the results indicate that the overall global warming potential (GWP) is reduced by 64% after accounting for all the life cycle stages of the carbon capture process. Furthermore, 75% of the GWP is due to direct emissions from the plant in the CCS scenario. More interestingly, most of the remaining GWP is due to natural gas production cycle emissions. Although the emissions related to natural gas production are considered in scenarios with and without CCS, the increase in the emissions in CCS scenarios is due to increased fuel consumption.

Furthermore, the study shows that $CO_2$ storage and transportation only account for less than 3% of the GWP. Petrescu et al. (2017) conducted an LCA on pulverized coal power plants with carbon capture and storage [23]. The study shows that the percentage contribution of $CO_2$ storage and transportation on GWP impacts is 14%.

There are non-GHG-related environmental impacts that can result from carbon capture, although it can reduce the GWP significantly. In NGCC power plants, $SO_2$ is reduced from 3.1 mg/ kWh to 0.0005 mg/ kWh after being integrated with the MEA carbon capture system [88]. However, Korre et al. (2010) show that carbon using MEA chemical absorption can increase the acidification potential of the overall process by 20% [89] in a coal combustion plant. Furthermore, the acidification potential (AP) may increase up to 43% in natural gas power plants. The main reason behind the increase of AP is that a carbon capture process increases emissions of $NH_3$ and MEA.

Furthermore, the increase in fuel production and chemical production also increases the AP. More interestingly, coal power plants have a lesser impact on AP compared to NG power generation plants. This is due to the considerable loss of $SO_2$ emissions in coal combustion due to the flue gas desulphurization (FGD) that is used especially for the carbon capture process [90]. In addition, Singh et al. indicated that the carbon capture process increases toxicity by more than 100% due to increased fuel consumption and direct emissions of compounds such as formaldehyde and MEA [87].

Although storing captured $CO_2$ in geological or offshore storage sites can retain $CO_2$ permanently, the utilization of $CO_2$ may not. The production of $CO_2$-derived polymers and using $CO_2$ for yield boosting greenhouses have lower relative climate benefits, as the majority of $CO_2$ is released into the atmosphere over a short period of time [91]. $CO_2$-derived fuels have medium climate benefits due to combustion releasing $CO_2$ into the atmosphere [91]. $CO_2$-cured concrete aggregates and building materials are highly beneficial to the atmosphere as they can keep most of the $CO_2$ in the building materials for a long time [91]. Therefore, the fate of the captured $CO_2$ must be considered when studying the environmental impact of the carbon capture process.

5.1.1. Evaluation of GHG Emission Reduction of Building-Level Carbon Capture Technologies

Liyanage (2021) [92] evaluated the life cycle GHG emissions of building-level carbon capture technologies. The study presented a reduction of life cycle GHG emissions when using the KOH-based building-level carbon capture technology discussed in Section 4 and when adopting MEA-based chemical absorption technology. The life cycle GHG emissions

were evaluated when the carbon capture systems were integrated into natural gas DHW heating systems in office and multifamily building buildings located in eight provinces in Canada. The building energy loads were determined by the CANQUEST energy modeling software. The $CO_2$ capture rate, energy consumption, raw material requirement, heat recovery rate, and by-product generation were determined using literature-based average performance data, manufacturer data, experimentation data, and energy and mass conservation principles. The life cycle GHG emissions was determined by using Simapro software and the TRACI 2.0 impact assessment method. The life cycle system boundary was cradle to gate, and the functional unit was generating 1 GJ of thermal energy per year.

The study [92] indicated that integrating an MEA-based carbon capture system may reduce the life cycle of GHG emissions by 40% to 54%. Although the study assumed 90% carbon recovery in MEA systems, a significantly lower GHG emission reduction was observed due to the various GHG emitting processes during the life cycle of the carbon capture process. Specifically, the study emphasized that the MEA-based carbon capture process increased natural gas combustion by over 20% compared to conventional heating systems. Natural gas production significantly contributes to GHG emissions, which increase the overall life cycle of the GHG emissions from the MEA-based carbon capture process. In addition, the electricity consumption of the carbon capture process contributes 1–24% of the life cycle GHG emissions. A higher contribution was shown in locations that depend on fossil fuel to generate electricity.

The study [92] showed that the KOH-based building-level carbon capture system might increase the life cycle GHG emissions due to the significant energy consumption of raw material (KOH) production. However, the study also indicated that the KOH-based carbon capture system might reduce the overall life cycle GHG emissions when accounting for the emission avoided due to by-product ($K_2CO_3$) production. The KOH-based system can reduce GHG emissions by 17–40% in multifamily buildings and 17–20% in office buildings when considering that the production of $K_2CO_3$ was avoided. In the KOH-based system, GHG emission reduction potentially changes with the building's energy load due to the material handling capacity limitation and the heat recovery rate (The KOH-based carbon capture system has a monthly KOH storage capacity of 200 kg). Furthermore, the study [92] emphasized that the majority of the GHG emission reductions in the office building were contributed by the heat recovery system in the KOH-based system.

Apart from that, this literature review extended the case study from that study [92] by considering a long-term care facility and a university. The study adopted the same methodology developed by Liyanage (2021) [92] and Liyanage (2020) [61,62] to estimate the life cycle GHG emissions. More details regarding the building simulation, including the building sizes and locations, can be found in the Table A1 in Appendix A. Figure 6 shows the percentage reduction of life cycle GHG emissions of each building when integrated with the KOH-based carbon capture system.

### 5.2. Economic Costs and Benefits

The literature review shows a lack of knowledge in terms of the cost estimation of most carbon separation technologies. Rubin et al. have conducted a comprehensive review on the costs of carbon capture and storage [93] technologies that are applied for supercritical pulverized coal power plants (SCPC), natural gas combined cycle power plants (NGCC), and integrated gasification combined cycle power plants (IGCC). The study shows that NGCC with carbon capture would increase the levelized cost of electricity (LCE) by 26%. The cost of the avoided $CO_2$ could be 58–121 USD/t $CO_2$ without storage or utilization. Furthermore, when the captured $CO_2$ is utilized for enhanced oil recovery, the total cost of the avoided $CO_2$ is reduced to 10–112 USD/t $CO_2$.

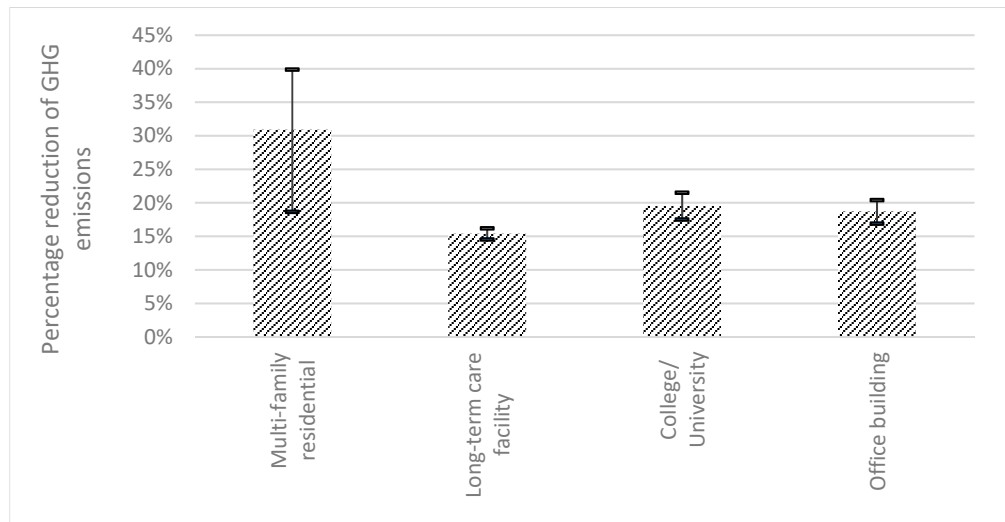

**Figure 6.** Life cycle GHG emission reduction of KOH-based building-level carbon capture technology.

The operational costs can be increased significantly by installing carbon-capturing systems. The operational cost of using carbon capture consists of the cost of chemicals, electricity consumption for auxiliary equipment, and fuel to generate heat for the building and solvent regeneration. The considerable increase of the operational energy due to the building-level carbon separation and transportation would be a barrier when integrating carbon capture systems in building heating systems. However, other possibilities can reduce the operational cost of carbon capture systems. Carbon taxes for fuels could be reduced, as integrating carbon capture reduces the GHG emissions from fuel combustion. Furthermore, there are carbon trading mechanisms such as "Cap-and-trade" [94] that have been implemented in provinces such as Nova Scotia [95]. Currently, only industries that produce more than 25,000 tons of GHG per year participate in this program. In this method, when GHG emissions are higher than the cap, the participants must purchase emission allowances or carbon offsets equal to the exceeded emissions. Conversely, if participants produce fewer emissions than the cap, they can sell their unused allowances. These programs have been implemented in the USA for residential and commercial buildings to promote a low carbon economy [96]. This method would help residents sell emission allowances that further reduce the operational cost of carbon capture systems [96].

Past literature shows that separating $CO_2$ and the utilization process always increases the energy generation process cost, despite the revenue generated from utilizing $CO_2$ and the reduction of tax. However, the cost of converting $CO_2$ into a different product during the carbon process is overlooked in studies. Furthermore, heating systems may be able to provide a continuous supply of $CO_2$ as a feedstock. Therefore, a building-level carbon capture process may create opportunities for production processes that use $CO_2$. In addition, converting $CO_2$ into a valuable product during the capturing process may have the potential to reduce total operational costs, as it does not require processes such as the compression, liquefaction, purification, and transportation of $CO_2$.

Table 6 shows the percentage increase of capital costs after integrating the carbon capture system [93]. This indicates that integrating carbon capture processes into NGCC power plants has the highest percentage increase of the capital cost. However, the capital cost per unit of power required for an NGCC power plant is substantially less than the capital cost per unit of power required for SCPC power plants [97]. Therefore, the cost increment percentage does not indicate that installing carbon capture systems is more costly for natural gas combined-cycle power plants.

**Table 6.** Percentage increase of capital cost of integrating carbon capture in power plants [93].

| Power Plant Type | % Increase in Capital Cost |
|---|---|
| Supercritical pulverized coal power plant | 58–91 |
| Natural gas combined cycle power plant | 76–121 |
| Integrated gasification combined cycle power plant | 30–47 |

Table 6 shows that the capital costs of power generation plants can be increased by up to 121% after integrating a carbon capture system. A power generation plant is a complex system, which includes various components such as boilers, turbines, heat exchangers, and generators. Therefore, if integrating carbon capture systems increases the capital cost by 121% in such a complex process, integrating carbon capture at the building scale may substantially increase capital costs. Therefore, the capital cost of the carbon capture is a very important factor that determines the potential of a building-level carbon capture system compared to its competitors. On the other hand, a carbon capture system can be designed on a smaller scale that only reduces part of the building-scale emissions. Such a solution may reduce the capital cost, although it compromises the GHG emission reduction potential.

Integrating a carbon capture system may, however, increase the job opportunities in small industries at the community scale. Specifically, emerging $CO_2$ utilization technologies such as $CO_2$-cured concrete and $CO_2$-derived fuels have a considerable potential to utilize more than 1 Gt of $CO_2$ per year. It may be more economical to supply the required amount of $CO_2$ from the nearby buildings when considering the significant transportation costs incurred when $CO_2$ is transported from long distances [93]. In contrast, the cost of the $CO_2$ captured from the buildings may be higher than the $CO_2$ derived from commercial industries, as the production scale is low in buildings. Therefore, future studies must be conducted to investigate the economic justification of the cost of $CO_2$ captured from building heating systems by considering the demand and the existing pathways of acquiring $CO_2$ and by-products in industrial applications.

Evaluation of Economic Cost and Benefits of Building-Level Carbon Capture Technologies

This section presents the results obtained from the life cycle cost evaluation of building-level carbon capture technologies. The case study explained in Section 5.1.1 was used to evaluate the life cycle costs. The methodology is explained as follows.

The life cycle cost was evaluated using the equation given below. Equation (2) was derived by study [62] based on the life cycle cost manual for the federal energy management program [98].

$$C_{LCC} = C_{AQC} - C_{RES} + C_{FC} + C_{OM} - C_{RG}, \tag{2}$$

where $C_{LCC}$ = the life cycle cost of the heating system integrated with the carbon capture system, $C_{AQC}$ = the investment cost of the system, $C_{RES}$ = present value of the residual cost of the system, $C_{FC}$ = present value of the fuel and raw material costs, $C_{OM}$ = present value of the operational and management costs, and $C_{RG}$ = revenue generated from selling by-products.

The equipment cost of the KOH-based system was given in the reference [62]. The raw material cost, by-product selling price, and operational and management costs were obtained from manufacturer consultations as recorded in reference [62]. Furthermore, provincial natural gas and electricity costs were also available in reference [62].

However, the investment costs of an MEA-based system had to be estimated, as they were not available for the building scale. Therefore, Equation (3) was used to estimate the bare erected cost of an MEA-based system [20]. It considers economies of scale when accounting for the capital costs of equipment. The bear erected costs and reference flows of the components were determined using reference [99]. Furthermore, the total investment cost consists of general, instrumentation, electric, and piping costs. These cost figures were calculated as a percentage of bear erected cost [100]. The raw material required for an

MEA-based system was estimated using IECM software [101,102]. The operational and maintenance costs were estimated as a percentage of the equipment cost (2.5%).

$$BEC = C_0 \left( \frac{Q}{Q_0} \right)^f, \tag{3}$$

where $BEC$ = bear erected cost of the component, $C_0$ = bear erected cost of the reference component, $Q_0$ = size of the reference component, and $Q$ = size of the component

In addition to the life cycle costs, the study also estimated the selling prices of the by-products to achieve a given payback period using Equation (4).

$$DPP = \ln \left( \frac{1}{1 - \frac{IC \times r}{E}} \right) \bigg/ \ln(1 + r), \tag{4}$$

where $DPP$ is discounted payback period, $IC$ is the investment cost, $E$ is the annual savings, and $r$ is the discount factor.

Figures 7–10 show the life cycle cost and life cycle cost per reduction of 1 kg of GHG emissions in each building scenario. The figures indicate that MEA-based systems may significantly increase life cycle costs compared to conventional natural gas heating systems. The lifecycle costs per reduction of 1 kg of $CO_2$ by the MEA-based system are CAD 4.80–7.00 in a multifamily residential building, CAD 0.50–0.80 in a long-term residential building, CAD 1.00–1.50 in a college, and CAD 0.90–1.35 in an office building. This is due to the significant increase in the capital and operational costs of MEA based systems. The study revealed that KOH-based systems perform considerably better than MEA-based carbon capture systems, considering the life cycle cost per reduction of 1 kg of $CO_2$. In addition, it can be observed that the lifecycle cost per reduction of 1 kg of $CO_2$ in KOH-based systems in commercial buildings can also be negative. It was due to the considerable operational cost reduction by selling by-products and the natural gas savings by the heat recovery systems.

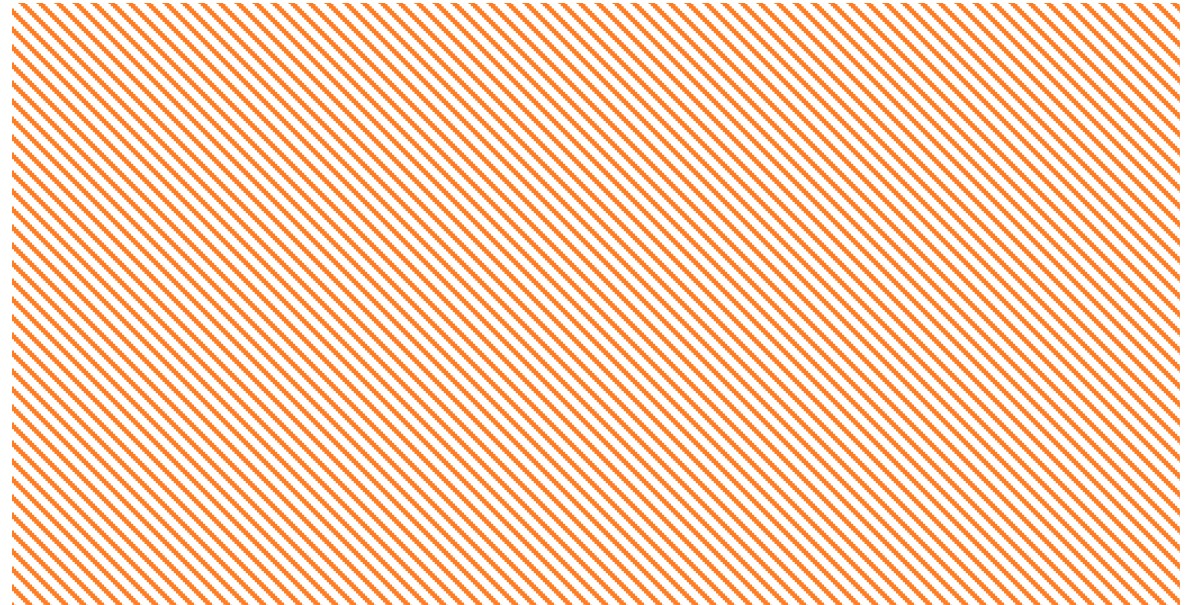

**Figure 7.** Life cycle costs and life cycle cost per reduction of 1 kg $CO_2$ eq of life cycle GHG emissions in multifamily residential building.

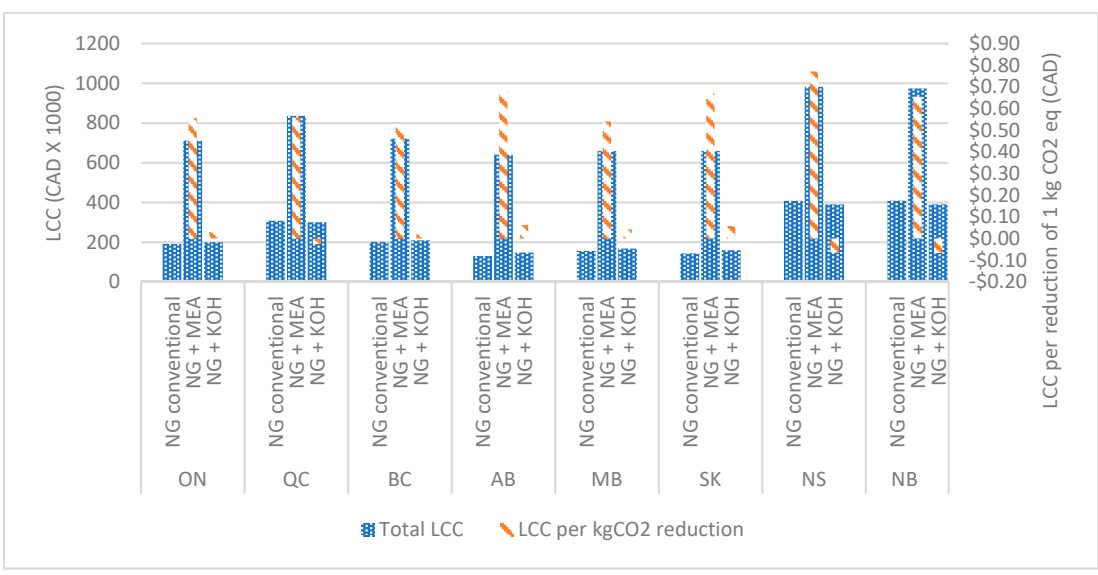

**Figure 8.** Life cycle costs and life cycle cost per reduction of 1 kg $CO_2$ eq of life cycle GHG emissions in long-term care facility.

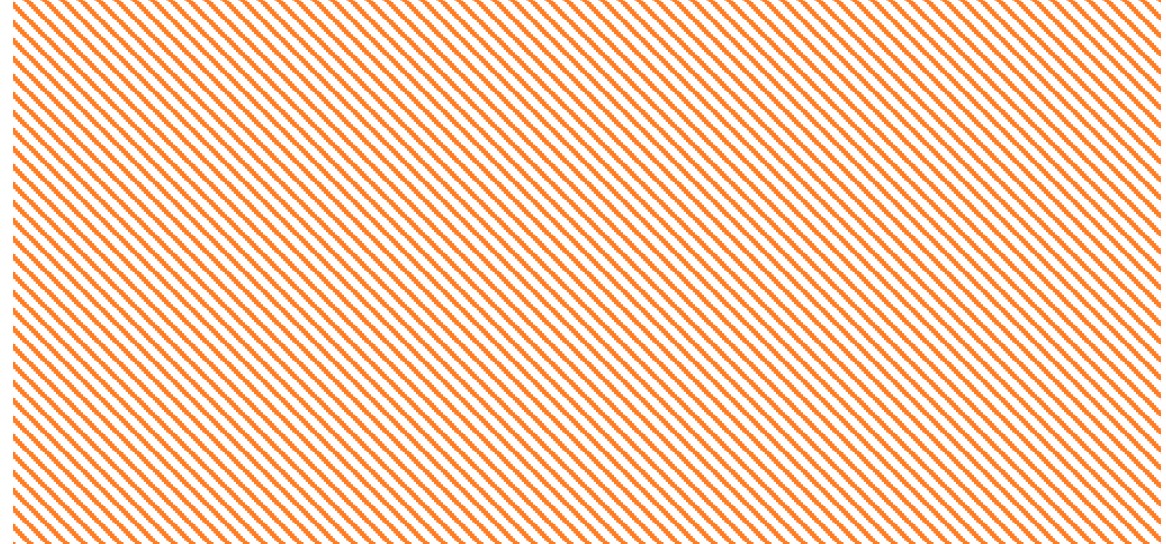

**Figure 9.** Life cycle costs and life cycle cost per reduction of 1 kg $CO_2$ eq of life cycle GHG emissions in long-term care facility.

Figure 11 shows the $CO_2$ selling prices that are required to achieve the payback periods when MEA-based systems are integrated into the natural gas heating systems of buildings. The economic value of $CO_2$ is USD 36/t$CO_2$ (CAD 0.043/kg$CO_2$), which was based on the price of $CO_2$ that can be paid for the enhanced oil recovery EOR (2% of the oil price in 2015). However, this study shows that the price of $CO_2$ must be substantially increased in order to pay back the investment cost within the lifetime of the system (20 years). The highest price required to be able to pay back the investment within the lifetime of the system was observed in the multifamily residential building due to the lower generation of $CO_2$, while the lowest price of $CO_2$ was observed from the long-term care facility, which has the highest annual energy consumption because of water heating.

Figure 12 shows the selling prices of $K_2CO_3$ in KOH-based scenarios for different payback periods. The manufacturer of the KOH-based building-level carbon capture system revealed that $K_2CO_3$ selling prices vary from CAD 1.7–4.0 per 1kg [62]. The study considered average the selling price to evaluate the operational costs of building-level carbon capture technologies. However, the results show that the by-product selling prices must be higher than the market prices in order to pay back the investment over the

lifetime of the system when the KOH-based system is installed in multifamily residential buildings. When installing the KOH-based system in other buildings, the results show that the investment cost can be paid back during the lifetime of the system. However, the selling price of KOH must be higher than the average selling price of $K_2CO_3$.

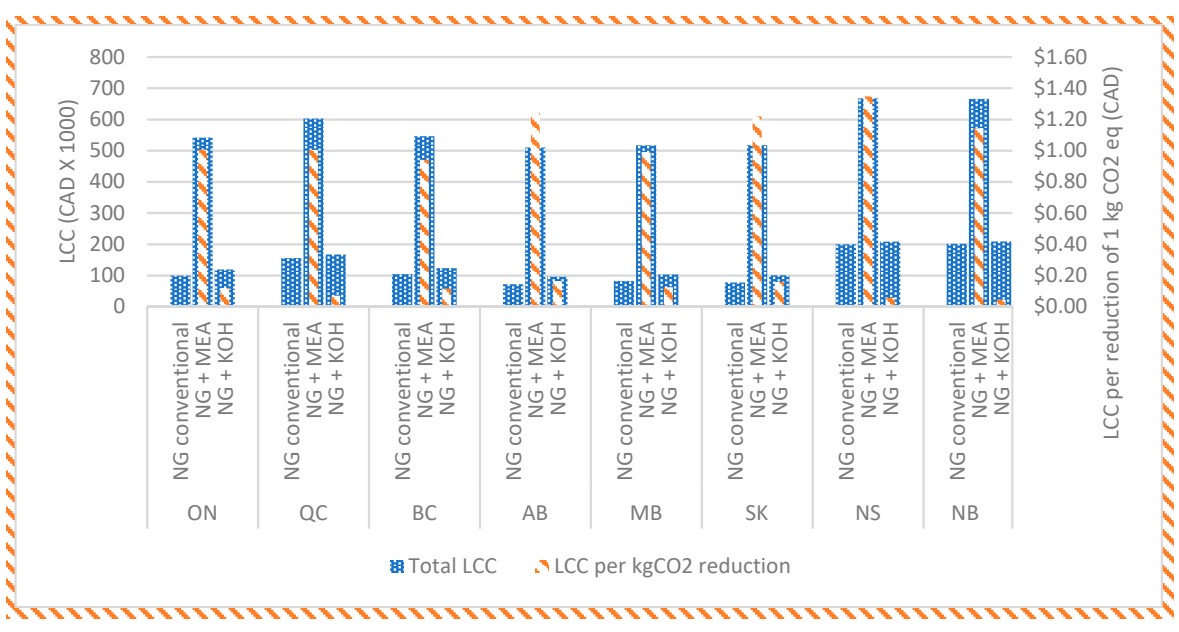

**Figure 10.** Life cycle costs and life cycle cost per reduction of 1 kg $CO_2$ eq of life cycle GHG emissions in office building.

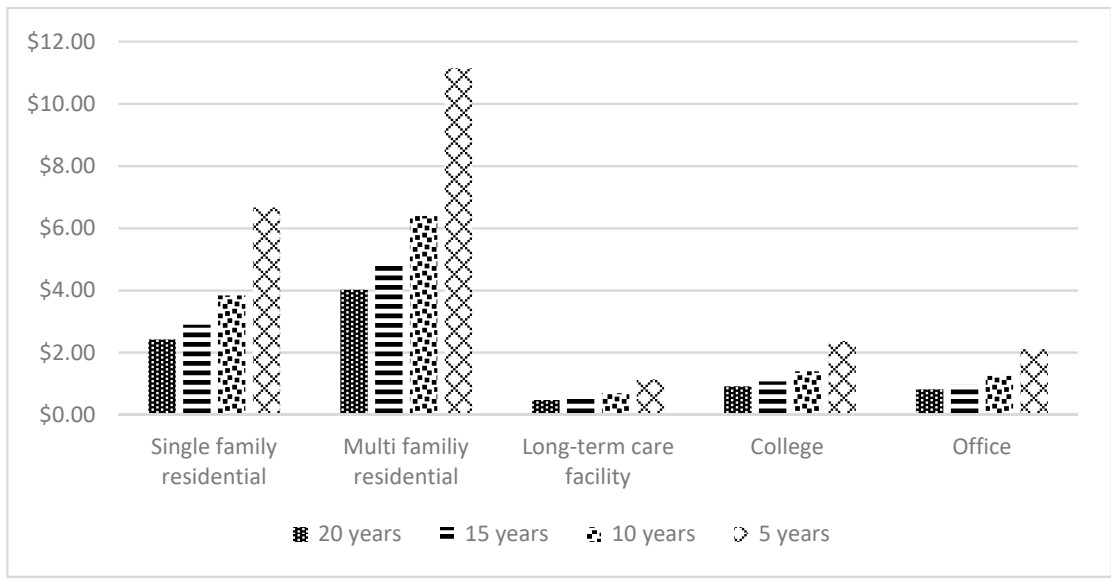

**Figure 11.** Payback period analysis of MEA-based carbon capture technology when integrated into natural gas building heating systems.

### 5.3. Social Acceptance

Understanding the social acceptability of the building-level CCUS is vital for successful implementation. There are many examples of introducing innovative technologies that fail or that are delayed due to failure to consider the opinions of key social actors [103]. The triangle of social acceptance concept that was developed by Wüstenhagen et al. and has been used in many studies to investigate the social acceptability of innovative technologies [104,105]. The social acceptance triangle consists of socio-political acceptance, market acceptance, and community acceptance [105]. The acceptance of technology and

policies by major social actors such as the general public and policymakers is known as socio-political acceptance. Market acceptance refers to the acceptance of the technology among consumers and technology investors. Community acceptance refers to the acceptance of the community stakeholders such as residents in regions where the development of the technology occurred.

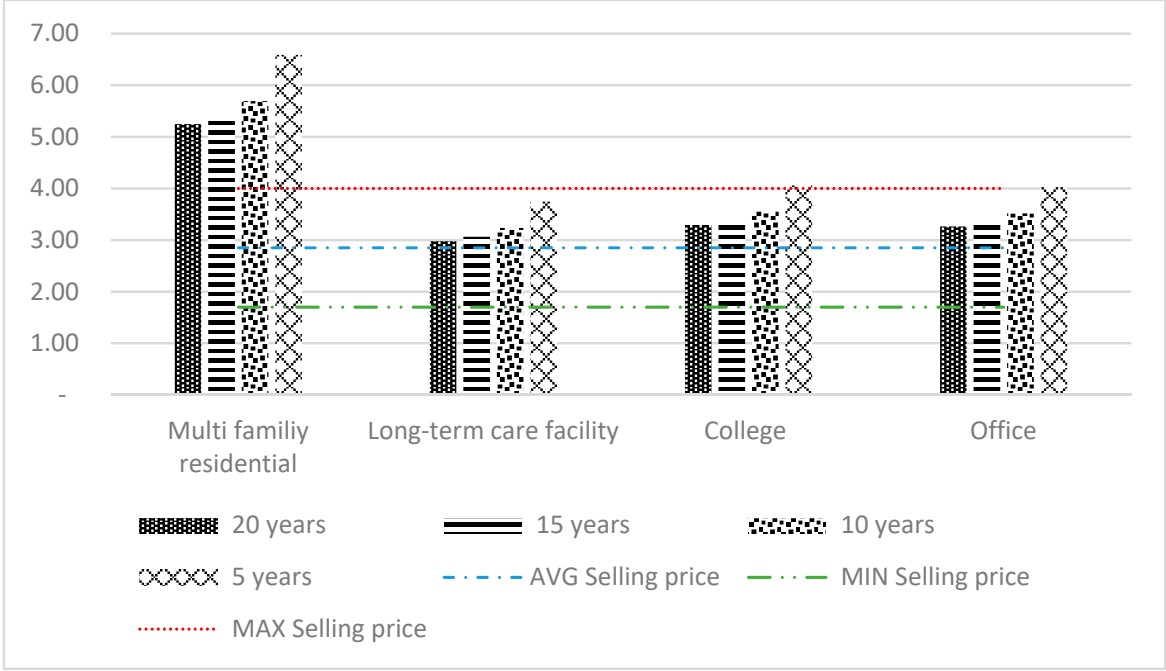

**Figure 12.** Payback period analysis of KOH-based building-level carbon capture technologies.

The general public, public authorities, and policymakers are the key actors in the socio-political context. Socio-political acceptance is an essential dimension of the social acceptance triangle. Social actors are responsible for institutionalizing frameworks such as procurement mechanisms and decision-making protocols, which may directly affect the market and community acceptance of innovative technologies [105]. The key driver of the socio-political acceptance among policymakers regarding building-level CCUS would be climate change mitigation. Political pressures such as reducing carbon footprint and achieving climate mitigation targets may drive public authorities towards innovative climate mitigation technologies such as building-level CCUS.

Building owners and utility providers are the major consumers and technology investors of the building-level CCUS technology. The acceptance of this technology by building owners may be negatively affected by the intense maintenance efforts related to the carbon capture process that may require significant technical expertise. In addition, the captured $CO_2$ and produced by-products must be transported from the buildings. These processes involve significant infrastructure and additional work requirements. Sound generation due to pumps, compressors, and other carbon capture equipment may also raise issues in the building environment. Therefore, introducing such a process may not be successful unless significant benefits for the building owners are realized. On the other hand, utility providers may be driven by the taxes and economic penalties related to the carbon footprint of their industries. The development of building-level CCUS will substantially reduce the increasing economic burdens and pressures put on natural gas utility providers.

Pockhrel et al. (2021) conducted a market assessment on building-level carbon capture technologies [63]. The study focused on assessing the perspective of building managers on implementing building-level carbon capture technologies. This is the only study found in the literature review on the social aspects of building-level car carbon capture technologies.

The study indicated that 50% of the participating non-residential building managers were interested in building-level carbon capture. The study also emphasized that there was significantly low interest in the residential sector. The lack of interest, however, was mainly due to the lack of evidence regarding accurate estimations of the economic benefits of building-level carbon capture technologies.

Wallquist et al. studied the public acceptance of CCS technology elements, including carbon capture, $CO_2$ transportation, and $CO_2$ storage [106]. The study found that the early stage of the CCS technology negatively affects the community's perceptions of the possible risks related to the technology. In addition, $CO_2$ transportation was considered the most important element, while most participants disliked having to live near $CO_2$ transportation pipelines due to the risk of leakage. Although pipeline $CO_2$ transportation may not be viable due to the smaller scale of applications in the building sector, the result shows the need to assess the risks behind the building-level $CO_2$ transportation process, which may otherwise negatively affect community acceptance. Jones et al. studied the lay perception of $CO_2$ utilization technologies [107]. The study revealed that the awareness of the $CO_2$ utilization technologies was low in the participants. However, the participants tended to support the concept of $CO_2$ utilization. Since building-level carbon capture can create a community-level carbon economy, it positively affects society by increasing job opportunities. It reduces the unemployment rate in the community and increases economic welfare, which is a positive social impact. The community will also have a chance to contribute to the economy while reducing the global warming potential incurred due to fossil fuel combustion.

The technology readiness level of building-level CCUS technology is low due to the lack of research and development. Public funding schemes and research program investments are important to encourage the development of technologies that are in their early technology readiness phases [105]. In addition, the academic community has an important role to play in supporting technology investors in the decision-making process [105]. It is necessary to evaluate the risks and benefits associated with different carbon capture technologies from environmental, economic, and social perspectives by considering future scenarios such as changes in technology maturity, the demand for by-products, and the development of carbon utilization technologies. The social acceptance of implementing building-level CCUS also depends on the ownership and the responsibility of maintaining the carbon capture system and transporting the captured $CO_2$ and its generated by-products. Therefore, developing stakeholder partnerships that include sharing the responsibilities, risks, and benefits of the building-level CCUS process is vital for successfully implementing the carbon capture process.

Table 7 summarizes the drivers and barriers of building-level CCUS implementation in an environmental, economic, and social context.

**Table 7.** Drivers and barriers of CCUS implementation in an environmental, economic, and social context.

| Context | Drivers | Barriers |
|---|---|---|
| *Environment* | Substantial reduction of GHGs from building heating system<br>Utilization of captured $CO_2$ may avoid the production of a different product | Increase of non-GHG environmental impacts<br><br>The captured $CO_2$ may release into the atmosphere in some utilization technologies<br>Higher embodied emissions |

**Table 7.** *Cont.*

| Context | Drivers | Barriers |
|---|---|---|
| *Economic* | $CO_2$ utilization sector is emerging | The investment cost of building heating systems can be increased |
| | The fuel supply infrastructure and current heating systems will not be changed. | The operational cost can be increased |
| | CCUS implementation may reduce the taxes and economic penalties related to the carbon footprint of the natural gas utility providers | Higher capital cost for $CO_2$ transportation |
| | Carbon tax rebates can be implemented on building owners | |
| *Social acceptance* | the climate change mitigation due to CCUS implementation may attract public authorities and policymakers | The implementation of CCUS may increase the efforts of maintenance and management of building heating systems |
| | The carbon economy may increase investment and job opportunities in the region | Operating the carbon separation systems, including handling chemicals, may require substantial technical skills |
| | | Chemical emissions in the building sector must be highly regulated compared to the industrial context |

## 6. Roadmap for the Feasibility Assessment of Building-Level CCUS Technologies

The literature review indicated that the feasibility of building-level CCUS depends on various technical, economic, environmental, and social factors. Therefore, this study developed a roadmap for the feasibility assessment of building-level CCUS based on the literature review findings. The road map is elaborated below and is illustrated in Figure 13.

- *Technical performance assessment of building-level carbon capture technologies*: Operation parameters such as $CO_2$ purity, reliability, and adaptability must be evaluated, and the threshold performance levels must be established to investigate the technical compliance of carbon capture technologies at the building scale, as these parameters may differ at the building scale compared to fossil power plants.
- *Environmental and economic performance assessment*: The literature review shows that the cost of building-level CCUS systems will be notably high although they can reduce GHG emissions. Therefore, environmental and economic performance must be evaluated and compared against the other alternative GHG mitigation technologies used in the building heating systems. In addition, it is essential to consider techniques such as multi-criteria decision-making for the comparative assessment, as the economic and environmental impacts of CCUS conflict with each other.
- *Future dynamics and potential changes in the macro-environment*: External factors such as variations in the demand for by-products and the development of new carbon utilization technologies must be considered when establishing the feasibility of building-level CCUS technologies. In addition, the effect on the triple bottom line sustainability of building-level carbon capture caused by changes in the macro-environment, such as social acceptance, the health risks related to the chemicals used, economic state, technology improvements, political involvements, carbon taxation, and carbon pricing, must be studied.
- *Investigate the supply chain and stakeholder partnership*: The supply chain of building-level carbon capture technologies must be investigated. The feasibility of carbon capture technologies depends on a properly established supply chain. In addition, the stakeholder partnerships within the complete carbon capture process must be thoroughly studied. Sharing responsibilities such as maintenance, infrastructure development, and by-product transportation must be considered when evaluating the effect of different stakeholder partnerships.

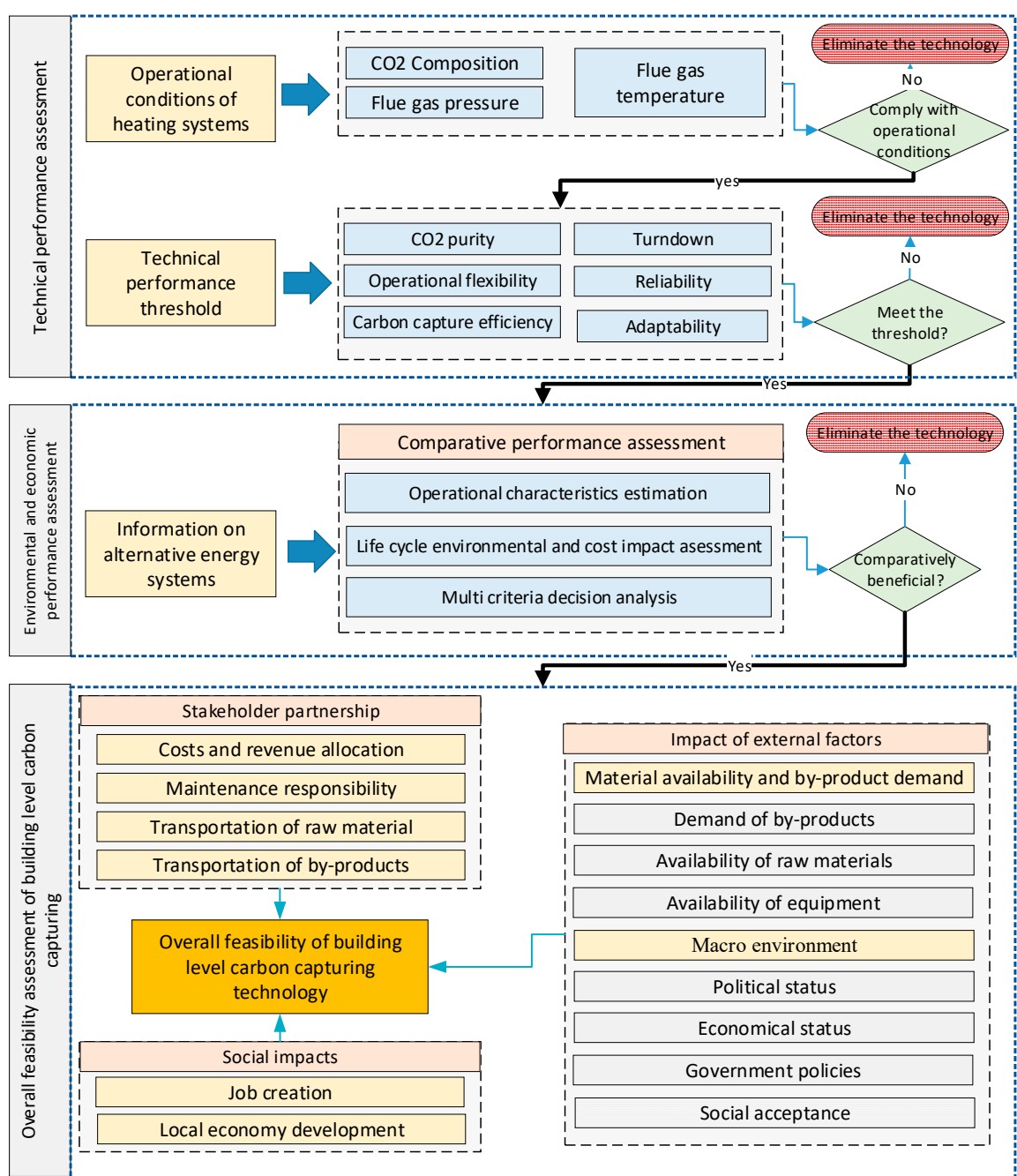

**Figure 13.** Roadmap for feasibility assessment of building-level CCUS technologies.

## 7. Conclusions

The main purpose of this paper was to provide guidance to assess the technical and triple bottom line sustainability of integrating carbon capture, storage, and utilization in building heating systems to reduce the GHG emissions. A critical literature review was conducted to investigate the potential of integration of carbon capture, utilization, and storage (CCUS) in natural gas building heating systems. The review focused on adopting the value chain of the carbon capture process used in fossil fuel combustion power generation facilities. The operational conditions required for the optimum operation of the $CO_2$ separation technologies were further investigated. In addition, operational parameters such as the operational flexibility, turndown, and reliability of $CO_2$ separation technologies were reviewed. This helped to identify suitable carbon separation technologies that can function in the building context. The study also discussed the possible pathways

of $CO_2$ transportation in building scale applications. In addition, the study conducted a preliminary assessment on the life cycle GHG emissions, life cycle costs, and pay back periods of CCUS technologies that can be adapted at the building scale. Finally, the study summarized the potential drivers and barriers of installing carbon capture technologies at the building scale and provided a road map for further research.

Among the three main carbon capture technologies, the post-combustion technology can be used with NG building heating systems. Therefore, the carbon separation technologies used in post-combustion technologies were considered. The study revealed that the membrane separation technology is more favorable for building heating systems based on operational parameters such as turndown, reliability, and adaptability. However, membrane separation and adsorption technologies require flue gas recirculation due to the low $CO_2$ concentration in building heating systems. It may require significant changes in the building heating systems. Chemical absorption and adsorption technologies have moderate performance over operational parameters. Moreover, the study indicates that the chemical absorption technologies can be used directly with building heating systems without modifications to the combustion systems.

The study revealed that adopting CCUS technologies can reduce the life cycle GHG emissions of natural gas-based building heating systems. Currently available building-level CCUS systems have a relatively lower reduction in GHG emissions compared to when the CCUS technologies used in the power generation sector are adopted into the building sector. Although the carbon capture process reduces GHG emissions, there can be adverse environmental impacts, such as increased human toxicity and acidification potential, from carbon capture systems. The economic analysis revealed that adopting the CCUS technologies used in the power generation sector into the building sector may require substantial investment compared to current building-level technologies. This was identified as one of the main barriers. Currently available building-level CCUS technologies can recover the investment costs in commercial and institutional buildings through the revenue generated from by-products sold at the current market price range. Furthermore, the life cycle costs may be reduced with energy-efficient carbon capture technologies and policy-level involvement such as tax reductions and the introduction of carbon credits. The review on the studies of the social aspects of CCUS technologies revealed that building-level CCUS technologies are gaining interest in the commercial and institutional building sectors. However, social acceptance may require more accurate estimations of the technological, economic, and environmental performance of building-level CCUS technologies.

The study showed that implementing CCUS at the building level has great potential as a climate change mitigation method when considering economic, environmental, and social aspects. However, there are barriers to implementing CCUS at the building level that would affect the commercialization. These must be thoroughly studied along with the solutions for the successful commercialization of building-level CCUS. In addition, stakeholder partnerships, risk and benefit sharing mechanisms, and the ownership structure of the building-level CCUS technology should be investigated to understand the applicability of building-level CCUS in terms of practical implementation. The findings emphasized the need for further study on integrating carbon capture technology in building-level heating systems. The information gathered in this study helps researchers, policymakers, building owners, and developers to assess the technical and triple bottom line sustainability of building-level carbon capture.

**Author Contributions:** Conceptualization, D.R.L. and H.K.; formal analysis, D.R.L.; investigation, D.R.L.; writing—original draft preparation, D.R.L.; writing—review and editing, H.K. and G.C.-S.; supervision, R.S. and K.H.; project administration, K.H. All authors have read and agreed to the published version of the manuscript.

**Funding:** This research was funded by the Natural Sciences and Engineering Research Council of Canada (NSERC) through a Collaborative Research and Development Grant (CRD).

**Institutional Review Board Statement:** Not applicable.

**Informed Consent Statement:** Not applicable.

**Acknowledgments:** The authors gratefully acknowledge the support provided by the Natural Sciences and Engineering Research Council of Canada (NSERC).

**Conflicts of Interest:** The authors declare no conflict of interest. The funders had no role in the design of the study; in the collection, analyses, or interpretation of data; in the writing of the manuscript; or in the decision to publish the results.

## Appendix A

**Table A1.** Building model information for multifamily residential, commercial, and institutional buildings.

| *Information Category* | *Information* |
|---|---|
| *Building types and number of stories* | • Medium-rise multifamily residential building—5 story<br>• Long-term care facility (Nursing home)—Single story<br>• High-rise office building—8 story<br>• University/College—4 story |
| *Building locations* | • Ottawa, Ontario (ON)<br>• Montreal, Quebec (QC)<br>• Vancouver, British Columbia (BC)<br>• Calgary, Alberta (AB)<br>• Winnipeg, Manitoba (MB)<br>• Saskatoon, Saskatchewan (SK)<br>• Halifax, Nova Scotia (NS)<br>• Moncton, New Brunswick (NB) |
| *Building area* | Determined by changing the building area so that the capacity of the water heating systems equals the maximum capacity of the heating system that can be connected to the KOH-based MCCU system. |
| *Building footprint* | • Medium-rise multifamily residential building—Rectangular<br>• Long-term care facility (Nursing home)—"L" Shape<br>• High-rise office building—Rectangular<br>• University/ College—Rectangular |
| *Building operation schedule* | Default operational schedule defined in CANQUEST software |
| *Domestic water heating hourly profile* | Default water heating hourly profile defined in CANQUEST software |
| *Non-residential domestic water heating specifications* | Heater specifications<br><br>• Heater fuel—Natural gas<br>• Heater type—Storage<br>• Hot water use—Determined by the software<br>• Input rating—Determined by the software<br>• Thermal efficiency–0.8<br>Storage tank<br><br>• Tank capacity—Determined by the software<br>• Standby loss—1.46%<br>Water temperature<br><br>• Supply water temperature–43.3 (Default settings)<br>• Inlet water temperature equals the ground temperature |

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
