# Peer review of "Carbon Capture Systems for Building-Level Heating Systems—A Socio-Economic and Environmental Evaluation"

_sustainability, doi:10.3390/su131910681_

Round 1
Reviewer 1 Report
This article mainly carries on the related aspect analysis research through the literature retrieval method. The comments are as follows:
- The special name for the related research, as I understand it, is CCUS.
- This paper has a good literature review and analysis, but the analysis of the targeted problem “Carbon-capturing systems for building level heating systems” seems to be not in-depth enough, Such as lack of relevant theory, model or method.
- In the conclusion part, the summary of the research results is too rough to reflect the research value.
Author Response
Thank you very much for the comments. Please see the response attached below.

Reviewer 2 Report
The study discussed the energy consumption used in buildings linked it to environmental degradation. The study presented a number of remedial measures to mitigate GHG emissions through sustainable instruments. The study is well written and have achieved the stated objectives of the study. The only suggestion is to increase the size of the literature review and included more relevant and current literature to support your arguments.
Author Response

(The authors gave the same response as above.)

Reviewer 3 Report
Comments
For the development of this article a descriptive qualitative research of reality with quantitative nuances was applied when measured in economic values ​​(cost-benefit analysis) and financial (operational and maintenance costs). The option to use case studies represents, in general, the representation of the preferred strategy when questions such as "how" and "why" are asked, when the researcher has little control over events and when the focus is on phenomena inserted in the context of real life. In this case of carbon capture systems, the elements were collected through analyzed documents. Is coal heating economical? Coal, along with wood, was one of the most important fuels in ancient times. Coal ovens were in virtually every home. Could coal be an alternative today? Fossil coal, in which vegetable remains can still be recognized, with a fixed carbon content lower than that of hard coal, brown or black, burns with a lot of smoke, as do brown coal, lignite, lignite, lignite and 0 lignite. While brown coal is a soft coal, hard coal is one of the coals that is slightly more energy efficient than lignite, but has an equally poor CO2 balance. The environmental problem of mining is not as dramatic with hard coal as compared to lignite, as - at least in Europe - there is no particularly intensive open-pit mining here. The promotion of hard coal is not without problems either. On the one hand, there are special charcoal stoves for burning charcoal, specially designed for low temperatures and cannot always be burned with wood. Coal-operated heating systems are now very rare. Cost-effectively, coal is now comparatively expensive as a heating material - many ecologically less objectionable alternatives are also much cheaper and, moreover, more convenient than digging coal. What's the alternative? Biomass. Not only logs but also pellet stoves or pellet heating can be a good and much more economical alternative. Biomass is also ecologically harmless and is not linked to the presence of fossil energy reserves. For the conversion of a possible heating of old coal still existing in biomass - on pellets, wood chips or a combined firewood and wood chip heating - today there are a multitude of state subsidies, which can clearly reduce the conversion costs. But solar thermal heaters, infrared heating or combined heat and power plants can also be an alternative to the old coal heating.
These concerns are part of the perspectives of using the Clean Development Mechanism (CDM) of the Kyoto Protocol, which aims to increase the use of renewable sources, reduce the emission of gases that cause the greenhouse effect in the atmosphere and contribute to its mitigation effect through carbon attachment. The objective is to reward ecologically correct production and renew interest in the use of charcoal in the steel industry.
The numbers also show that of the millions of meters of charcoal – MDC (unit of measurement for charcoal that is equivalent to the amount of charcoal that fits in one cubic meter) consumed, the production of pig iron by independent producers and by integrated plants is responsible for the high consumption of this. The search for cleaner and more effective technologies and the use of by-products (tar and gases) from the carbonization process highlight the evolution of traditional artisanal ovens to vertical cylindrical ovens that emerge as more efficient and effective. The charcoal production process consists of the partial degradation of wood and, therefore, it is necessary to apply sufficiently controlled heat for this to occur.
The origin of this heat can be classified in two ways: a) the partial combustion system or internal source of energy, where 10% to 20% of the load's weight is sacrificed, and b) the system that uses an external source of energy from the use of electric heating, or, c) the burning of other fuels introducing heat into the load, whose process yield is, therefore, greater, since theoretically there is no sacrifice of part of the wood by total combustion. Although they are cheaper and easier to build, they have low gravimetric yields - yields depending on the weight of fired firewood - in charcoal with losses in the form of polluting smoke that can reach 50% of the carbon initially contained in the fired firewood and 75% in weight of that same firewood. Gravimetric charcoal yields in the range of 25% obtained in traditional ovens represent a significant economic loss and underutilization of carbonized firewood.
The steps following the construction of the kiln are: a) acquisition of raw material (in most cases the firewood comes from its own reforestation or from third parties, and in some cases of legal management, duly proven with tax documentation); b) preparation of the raw material for this step uses the labor of two people for the production in the ovens of the analyzed system and consists of cutting logs that can vary in size between 1.00 and 1.40 m in length, according to the disposition of the load inside the kiln and, mainly, based on the experience of the person responsible for the filling, also called kiln or burner; c) filling: each kiln may have, in some countries, capacity for 16 wood stereos for the production of about 8 to 10 m3 of charcoal, depending on the variation in humidity, wood quality and handling in the assembly of the load in a cycle of up to 10 days; d) carbonization (during the kiln lighting process, all the holes remain open for about two hours, when only the chimney is sealed, the "baianas" remaining open - holes opened in the oven dome to control air inlet and outlet smoke, for about 5-6 hours). The vents - channels built on the sides of the oven with the same function as the "baianas", remain open for about 40-80 hours, depending on the humidity of the wood or until a bluish smoke manifests, when everything is sealed, starting if, thus, the oven cooling process, which can last up to four days, and this step is concluded from the perception of a temperature that is bearable to human beings; e) removal, bagging and shipping: the door, the “baianas” and the chimney are opened, allowing the light to enter, making it possible for the kilns to work in the process of removing and bagging charcoal. After bagging, the packaging edges are sewn. The mounting of the load on the transport vehicle is done in such a way as to accommodate the greatest amount of sacks, thus maximizing the transported weight, without prejudice to transport safety, whose maximum height must be 4.40 m from the ground, but variable according to legislation that varies from country to country.
Suggestions
It should be highlighted in the article that the search for cleaner and more effective technologies and the use of by-products (tar and gases) from the carbonization process highlight the evolution of traditional kilns to vertical cylindrical kilns that emerge as more efficient and effective. The charcoal production process consists of the partial degradation of wood and, therefore, it is necessary to apply sufficiently controlled heat for this to occur. The origin of this heat can be classified in two ways: a) by the partial combustion system or internal source of energy, where 10% to 20% of the load weight is sacrificed, and b) the system that uses the external source of energy from the use of electric heating, or, c) the burning of other fuels introducing heat into the load, whose process yield is therefore greater, since theoretically there is no sacrifice of part of the wood by total combustion. Although they are cheaper and easier to build, they have low gravimetric yields - yields depending on the weight of fired firewood - in charcoal with losses in the form of polluting smoke that can reach 50% of the carbon initially contained in the fired firewood and 75% in weight of that same firewood. Gravimetric charcoal yields in the range of 25% traditionally obtained represent a significant economic loss and underutilization of carbonized firewood.
These systems unequivocally show their own technical feasibility. The capacities for the system to be used are calculated proportionally to each company's survey, in order to make the comparison between the systems in terms of production and invoicing capacity more visual. In this way, the analysis and conclusions are visual and easy to interpret. It is verified that in the comparison between the systems, the carbonization system in vertical cylindrical furnaces, in contrast to the traditional carbonization system, presents a substantial economy in raw material (+/- 25%), a reduction in operating cost (+ /- 13%), an increase in profit (+/- 151%) and the return on investment measured by the simple “pay-back”, which takes place in less than 1 ½ years.
The importance of charcoal in the production chain is indisputable, but the productive model in use is debatable. On the one hand, the traditional charcoal production system, considered archaic from the (a) social point of view, where workers are subject, in many cases, to work without a work card, without the right to legal benefits, in addition to the conditions unhealthy; (b) the poor economic return of the production model, due to the high losses resulting from the low efficiency of the process and (c) environmental, showing neglected preservation. On the other hand, a sustainable model that can balance (a) social development, through the valorization of labor; (b) economic, via optimized use of raw material, adding financial value to the process and (c) environmental, as it contributes to the preservation of natural resources, through the better use of products and by-products from planted forests and the elimination of generated pollution during the production process, signaling an important direction of evolution for the sector. Having demonstrated the technical and economic feasibility, it is concluded that the implantation of the carbonization system in vertical cylindrical furnaces is important for the sector and, consequently, for society as a whole.
The governments of several countries have to try to change the energy industry towards the development and implementation of renewable sources. At the same time, some people are sure that the future lies in the coal mining industry. They don't realize that in addition to being cheap and easy to use, this is a fossil fuel and has many disadvantages. Furthermore, its burning is one of the reasons that cause global warming. What are the disadvantages of coal? There is a lot of coal at a disadvantage, but in this article we see a familiarization with the safer ones. Some of them are related to damage to the environment, while others have a negative impact on the atmosphere and even on health. Unfortunately, there are countries that depend on the coal industry. They use it to satisfy all their needs, but they don't take into account that this is a fossil fuel. Overnight, this energy source will be deprived and, if countries do not have a backup plan, they could enter an energy crisis. Fortunately, there are currently wind turbines and solar panels that can help to avoid this problem. In addition, the big news is that some countries have realized the possible danger and are taking steps to prevent the catastrophe. Last year, the Secretary-General of the United Nations urged to cease construction of new coal facilities in 2020, "if humanity does not want to face total disaster."
Finally, the coal combustion process generates not only energy, but also dangerous and radioactive “coal ash”. As a rule, it covers the areas close to the Coal Plant and causes colossal damage to the environment, including plants, animals and small ecosystems. On the other hand, in 2008, Oak Ridge National Laboratory (https://www.ornl.gov/content/solving-big-problems) working for the US Department of Energy, conducted special surveys and the results were shocking. According to the report prepared by the scientists, coal power plants release about 100% more heat into the environment than atomic power plants. And we must not forget that the coal combustion process generates environmental hazards, e.g., heavy metals, mercury and nitrous oxide.
Although there is special carbon capture and storage technology, it cannot handle carbon dioxide which is one of the main disadvantages of coal. Furthermore, it cannot reduce environmental contaminants as harmful as methane. Although this chemical normally dissipates into the atmosphere, it can enter the sea or ocean. In this case, methane can kill many marine ecosystems in a very short period of time.
Along with environmental problems, methane gas can cause various diseases. It could be the reason for different neurological and cardiovascular diseases. Scientists have already proven that this dangerous gas develops epilepsy, claustrophobia and even pneumonia. Also, it can cause depression and memory loss. Today, many scientists are certain that coal mines cause irretrievable ruin to the ecosystem in which they are located. It means that hundreds of animals and plants are constantly dying from coal production operations. Coal pollution causes rapid devastation of forests, rivers and fields. At the same time, there can be extremely dangerous fires in the coal joints. Sometimes it's impossible to forget them. Due to all these disadvantages of coal mining, many people have left their homes to avoid the harmful influence of nearby coal mines.
It can cause health problems caused by coal emissions. For example, coal ash includes silica which is extremely dangerous to our health. That's why when a person breathes coal dust they can have diseases such as asthma and silicosis. Also, regular exposure to these dangerous elements can cause cancer.
However, one of the most horrible causes of coal ash is called the “black lung”. Unfortunately, this is an incurable disease and usually brings death. A person who has this disease dies of asphyxia. Many scientists believe that the carbon dioxide that is generated during coal combustion is a major contributor to global warming. They report that 65% of the carbon dioxide in the troposphere was released by coal plants. It is obvious that this harmful effect increases the greenhouse effect that causes various cataclysms and climate change. In addition to these disadvantages of Coal Power Plants, coal combustion releases sulfur dioxide. Acid rain kills everything along the way. Although modern coal plants emit about 40% less CO2, this is not enough to repair the damage done in the past.
Fortunately, governments in more and more countries are taking this information into account and taking steps to stop the use of coal energy. For example, Norway decided to withdraw investments from companies whose activities depend more than 30% on coal energy. The most recent accidents in the US include the large upper branch and the mine disaster. At the same time, there are more victims in developing countries. This causes several economic problems. For example, the US paid more than $46 billion in federal compensation to miners and their families between 1971 and 2018, with the exception of coal mines. We must remember the transport of coal. Normally, transportation systems pass through abundant mountains and valleys. However, building transport lines can destroy these landscapes and kill the animals that live there. We must not forget that vehicles that are used to transport and increase emissions that pollute the air, according to surveys over the past 25 years, US coal mining companies have reclaimed more than 2 million acres of mined land.
Unfortunately, many scientists and ecologists say these places will never be fully recovered.
Author Response

(The authors gave the same response as above.)

Round 2
Reviewer 1 Report
It is better now.